# Predicting nutrient content of ray-finned fishes using phylogenetic information

Bapu Vaitla[1], David Collar [2], Matthew R. Smith [3], Samuel S. Myers[3,4], Benjamin L. Rice[5] & Christopher D. Golden [1,3]

Human food and nutrition security is dependent on marine ecosystems threatened by overfishing, climate change, and other processes. The consequences on human nutritional status are uncertain, in part because current methods of analyzing fish nutrient content are expensive. Here, we evaluate the possibility of predicting nutrient content of ray-finned fishes using existing phylogenetic and life history information. We focus on nutrients for which fish are important sources: protein, total fat, omega-3 and omega-6 fatty acids, iron, zinc, vitamin A, vitamin B12, and vitamin D. Our results show that life history traits are weak predictors of species nutrient content, but phylogenetic relatedness is associated with similar nutrient profiles. Further, we develop a method for predicting the nutrient content of 7500+ species based on phylogenetic relationships to species with known nutrient content. Our approach is a cost-effective means for estimating potential changes in human nutrient intake associated with altered access to ray-finned fishes.

[1] Department of Nutrition, Harvard TH Chan School of Public Health, Harvard University, 665 Huntington Ave, Boston, MA 02115, USA. [2] Department of Organismal & Environmental Biology, Christopher Newport University, One Avenue of the Arts, Newport News, VA 23606, USA. [3] Department of Environmental Health, Harvard TH Chan School of Public Health, Harvard University, 677 Huntington Ave, Boston, MA 02115, USA. [4] Harvard University Center for the Environment, Harvard University, 26 Oxford St, 4th Floor, Cambridge, MA 02138, USA. [5] Department of Organismic and Evolutionary Biology, Harvard University, 26 Oxford St, Cambridge, MA 02138, USA. These authors contributed equally: Bapu Vaitla, David Collar. Correspondence and requests for materials should be addressed to B.V. (email: vaitla@hsph.harvard.edu)

Human activity is rapidly transforming marine ecosystems worldwide. Total catch appears to be declining in fisheries around the world, largely due to intense industrial pressure[1]. Climate change-driven increases in sea temperature threaten to alter the abundance and distribution of hundreds of economically and nutritionally important species[2–5], and are leading to mass coral bleaching[6]. Ocean acidification, caused by greater absorption of atmospheric carbon dioxide, is inflicting severe damage on biologically rich coral reef habitats[7]. Coastal development, pollution, and other anthropogenic forces are impacting nearly every marine ecosystem in the world[8].

The consequences of these changes for human health, and especially for nutrient intake, are likely to be severe because fish provide critical nutrients essential to human nutrition, including iron, zinc, vitamin A, vitamin B12, omega-3 and omega-6 fatty acids, and others[9–11]. In many societies, seafood is the foundation for healthy diets, and its decline presents a significant risk in destabilizing food and nutrition security[12]. The consumption of fish is associated with a wide range of health benefits, including the prevention of various non-communicable diseases and the promotion of cognitive development[13,14]. Ray-finned fish—the class *Actinopterygii*—are particularly important to low-income populations, comprising 80.6% of the total global tonnage of subsistence marine capture fish and seafood in 2010[15].

A lack of information about the nutrient composition of fish species, however, hampers quantification of the nutritional threat to human populations of reduced consumption of wild-harvested fish. Measuring nutrient content is expensive and, as a result, nutrient analyses rarely capture the full breadth of vitamins, minerals, and macronutrients relevant to nutrition. This is exacerbated by the need to collect multiple samples of each species to assess variability across individuals, as well across subpopulations of species. This information gap prevents the design of rational fisheries management strategies and nutritional interventions to optimize public health outcomes in the face of rapidly changing marine conditions.

In this paper, we investigate the possibility that phylogenetic relatedness and life history information explain variation in the nutrient content of key fish species in the most commercially and nutritionally important class, *Actinopterygii* (ray-finned fishes). To our knowledge, this is the first time that such an approach has been explored. We then use the results of this analysis to develop a method for using shared phylogenetic history as a means of predicting the nutrient content of fish whose nutrient information has not yet been assessed. Such predictions are especially critical in regions of the world with known nutrient deficiencies—which are often the same areas where laboratory capacity is often limited. For fish specifically, recent years have seen an emerging desire among policymakers to design fisheries management and aquaculture development interventions with the specific goal of enhancing nutritional security. We note that these methods, if successful, could be used for a broad range of terrestrial animal and plant species as well—wild and domestic—as well as subspecies, varieties, and breeds.

Our results indicate that life history traits predict species nutrient content only weakly; larger datasets and/or use of missing-data imputation techniques may strengthen these models. We find, however, that phylogenetic relatedness is associated with similar nutrient profiles. This finding allows us to create a method for predicting the nutrient content of 7500+ *Actinopterygii* species based on phylogenetic relationships to species with known nutrient content.

## Results

### Interspecific variability in nutrient content and life history. We sourced nutrient content information for 371 species in the class

*Actinopterygii* (see Supplementary Table 1 for full list of species, and see Methods section and refs.[16–28] for data sources). Supplementary Table 2 summarizes the number of species with data, as well as the observed range by nutrient. The dataset includes 26 orders, 126 families, and 279 genera; about 42% of species are in the order *Perciformes* (perch-like fishes). The set of 371 species represents over half of all global capture fisheries by weight, with all 22 of the world's most harvested marine fish species included[29].

We used life history traits and phylogenetic relatedness to predict protein, total fat, omega-3 and omega-6 fatty acids, iron, zinc, vitamin A, vitamin B12, and vitamin D content in *Actinopterygii*. These nutrients are critical for human nutrition and are generally present in relatively high concentrations in seafood—although, as we describe below, not all species are rich sources of the all the above nutrients. As predictors of nutrient content, we initially considered six continuous and two categorical life history traits describing body size, trophic level, habitat characteristics, and geographic range (Table 1).

Summaries of nutrient content across species indicate that ray-finned fishes make substantial contributions to human nutrition, particularly for low-income coastal populations (Supplementary Tables 2, 3). Of species with available information for a given nutrient, 98% are sources or rich sources of protein, 94% rich sources of vitamin D, and 81% rich sources of vitamin B12. In addition, 13% of species are either sources or rich sources of iron, 14% sources or rich sources of zinc, and 10% sources or rich sources of Vitamin A (see Supplementary Table 1 for data by species and threshold designations of source and rich source by nutrient).

We see a great deal of variation across ray-finned fish species for which we have nutrient information (Supplementary Table 3). In particular, the ranges of minimum and maximum depth, habitat preferences, and latitudinal range are especially wide. Nearly 40% of fish are demersal, but significant fractions are benthopelagic, pelagic, or reef-associated. The chosen set is divided roughly equally into tropical, subtropical, and temperate species. As calculated by trophic level, the mean *Actinopterygii* species in this dataset is a carnivore, and some are apex predators, but many species are also herbivores, zooplanktivores, detrivores, or omnivores.

**Correlations among nutrients and life history variables.** We first examined the relationship between nutrient content and life history variables by estimating evolutionary correlations (i.e., the correlations between evolutionary changes inferred using observations among species and their phylogenetic relationships; see Methods section). For this analysis (and all regression models described below), we used the ray-finned fish phylogeny reconstructed in Rabosky et al.[30]. Due to sample size restrictions, we exclude omega-3 and omega-6 fatty acids from the correlation matrix below; however, correlations with fatty acids included are given in Supplementary Table 4.

We found the expected associations between pairs of life history variables (top left corner of Fig. 1, lightly shaded). Maximum depth, maximum fish length, and trophic level are moderately associated with each other, suggesting that larger fish tend to live at greater depths and consume a diet higher on the food chain. The *a* and *b* parameters describing species' mass-length scaling relationships are well-correlated to each other, but not to other life history traits.

Patterns are also apparent within the set of nutrients (bottom right corner of Fig. 1, darkly shaded). Fish with more protein tend to contain less Vitamin A and total fat. Fish that are high in total fat also tend to be good sources of Vitamins A, B12, and D. Fish high in iron also tend to be good sources of zinc and all included

**Table 1 Nutrient and life history traits used in the analysis***

| Nutrients | Description |
|---|---|
| Protein | "Serves as the major structural component of all cells of the body, and functions as enzymes, in membranes, as transport carriers, and as some hormones" |
| Total fat | "Energy source and, when found in foods, is a source of n-6 and n-3 polyunsaturated fatty acids" |
| Omega-6 polyunsaturated fatty acids (linoleic acid) | "Essential component of structural membrane lipids, involved with cell signaling, and precursor of eicosanoids. Required for normal skin function" |
| Omega-3 polyunsaturated fatty acids (α-linoleic acid) | "Involved with neurological development and growth. Precursor of eicosanoids" |
| Iron | "Component of hemoglobin and numerous enzymes; prevents microcytic hypochromic anemia" |
| Zinc | "Component of multiple enzymes and proteins; involved in the regulation of gene expression" |
| Vitamin A | "Required for normal vision, gene expression, reproduction, embryonic development and immune function" |
| Vitamin B12 | "Coenzyme in nucleic acid metabolism; prevents megaloblastic anemia" |
| Vitamin D | "Maintains serum calcium and phosphorus concentrations, and, in turn, bone health" |
| **Life history traits** | |
| Maximum length | Largest value ever reported for a given species, in cm (may not correspond to maximum length for subpopulation) |
| Trophic level | Weighted mean of the trophic level of the organisms that form the diet of the species; primary producers are assigned a value of one |
| Habitat | Categorical; bathydemersal, benthopelagic, demersal, pelagic, pelagic-neritic, pelagic-oceanic, reef-associated |
| Latitudinal range | Categorical; tropical, subtropical, temperate, boreal/austral, polar |
| Minimum depth | Minimum distance (m) below sea surface where species generally live |
| Maximum depth | Maximum distance (m) below sea surface where species generally live |
| $a$ length–weight parameter | Empirically determined scalar parameter of the function $W = aL^b$ |
| $b$ length–weight parameter | Empirically determined exponential parameter of the function $W = aL^b$ |

*Refs. [51-55]. All nutrient function descriptions taken from National Academies (2017); see ref. [50] for more details. See FishBase glossary at www.fishbase.org for detailed descriptions of life history traits

vitamins. Vitamin A and Vitamin D are positively associated, and the latter is correlated with Vitamin B12. Overall, this set of correlations suggests that specific sets of nutrients tend to cluster together. Given that nutritional analyses, due to cost considerations, tend to focus on a small set of nutrients, such observed correlations may be important for roughly inferring levels of unmeasured nutrients.

Finally, we see that life history parameters, controlling for phylogeny, vary in their association to key nutrients (top right of Fig. 1, unshaded). Fish living at greater oceanic depths contain more total fat and Vitamin A, but less zinc. Longer fish contain proportionally less zinc and Vitamin B12, but more Vitamin A; smaller fish, often overlooked in conservation, livelihoods, and nutrition programs, may be richer sources of key micronutrients[31].

These bivariate associations were tested in multiple regression models fit by phylogenetic least squares (PGLS) in a manner that accounts for phylogenetic signal in species residuals[16,32–34].

These phylogenetic regressions find that life history parameters are either insignificant predictors of nutrient content or have small effects (Table 2). Longer fish at higher trophic levels but closer to the surface contain more protein, although the coefficient magnitudes are relatively small: 100 cm greater length is associated with 0.69 g more protein; a one-unit higher trophic level is associated with 1.29 g more protein per 100 g of fish weight; and 100 meters greater (i.e., deeper) maximum depth is associated with 0.14 g less protein. Maximum depth is also a significant predictor for total fat, omega-3 and omega-6 fatty acids, and vitamin A, but regression coefficients are small in these cases as well. Other relationships are insignificant. Overall, regression models suggest that life history variables are generally poor predictors of fish nutrient content when accounting for phylogenetic dependence in the data.

**Predicting nutrient content of unmeasured species**. All nutrients exhibit significant phylogenetic signal; that is, covariance among species in nutrient content tends to be moderately to strongly associated with phylogenetic relatedness. Estimates of Pagel's λ vary[17,18] are highest (indicating strong phylogenetic signal) in zinc, vitamin D, and vitamin A, intermediate in total fat, omega-3 and omega-6 fatty acids, and vitamin B12, and weaker in protein and iron (Table 3).

The phylogenetic structuring of nutrient content can be seen in the heatmap of Fig. 2, which shows some clustering of nutrient values among closely related species, although the location of these clusters varies depending on nutrient. For example, various species of the genus *Thunnus* contain high levels of protein, as indicated by the dark red region near the top of the phylogeny in the first column of the heatmap. We also observe such clustering at close phylogenetic distance but across genera; for example, *Engraulis encrasicolus*, *Tribolodon hakoensis*, *Misgurnus anguilli-caudatus*, and *Anguilla japonicus* all fall within the top 10% of zinc levels in the database, as indicated by the dark red region near the bottom of the phylogeny in the fourth column of the heatmap; other nearby species also contain relatively large amounts of zinc.

Given our findings that nutrient content is weakly predicted by life history but tends to be phylogenetically structured, we developed a procedure for predicting nutrient content in unmeasured species based on their phylogenetic relationships to measured species and empirical estimates of phylogenetic signal (see Methods for details). We used a jackknifing procedure to determine that this method has acceptable accuracy. Observed nutrient values fell within the prediction 95% confidence intervals for at least 89.7% of cases across all variables (see Fig. 3, as well as Supplementary Fig. 1 for residual plots) and median deviations between predicted and observed values were within 40% of the among-species standard deviation. We note that the method can lead to wide confidence intervals when long phylogenetic branches separate unmeasured from measured species. This prediction method also compares favorably to predictions based on life history regression models; average accuracy and median

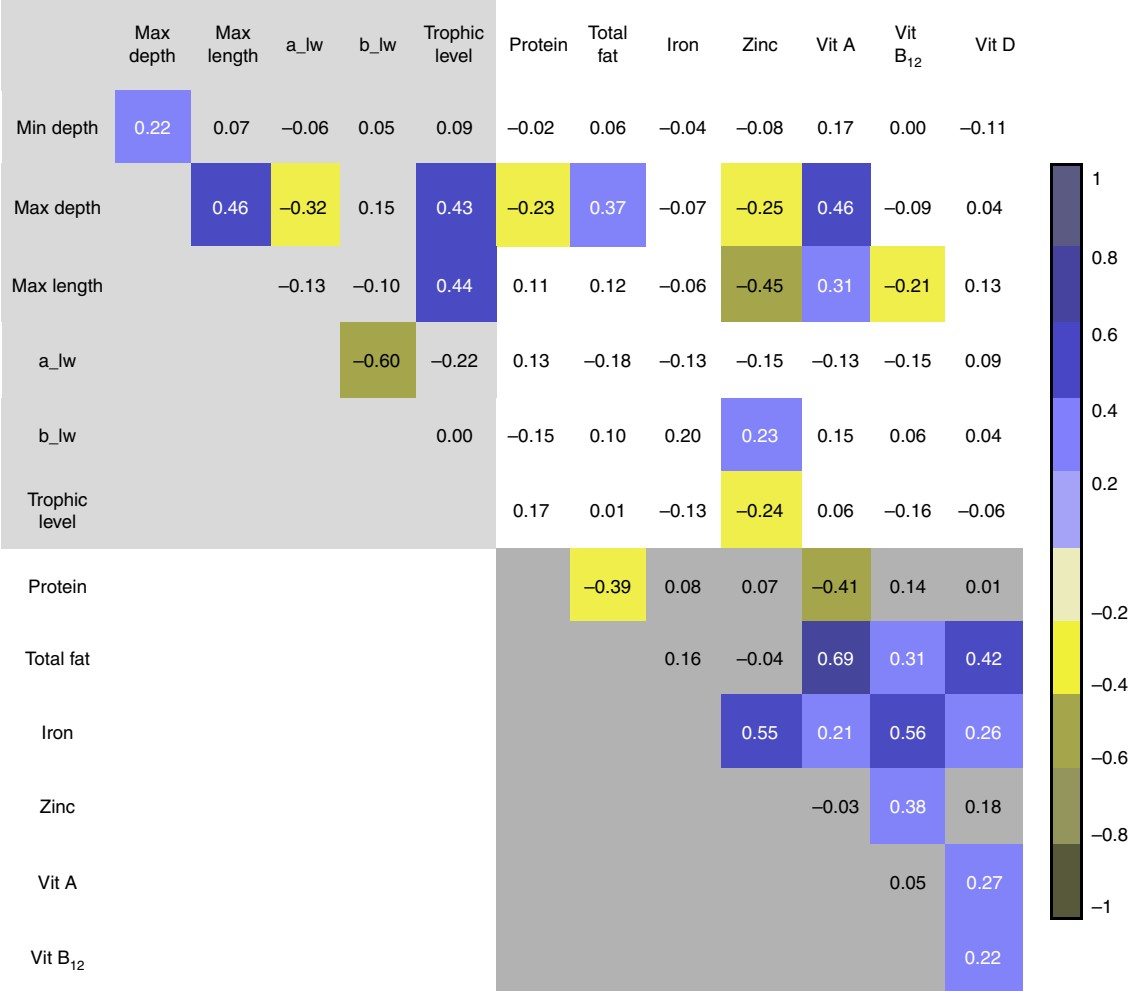

**Fig. 1** Evolutionary correlation matrix of key variables, log-transformed. Colored cells indicate correlations significant at $p < 0.1$ (Pearson's r). Lightly shaded top left corner shows correlations between life history variables; darkly shaded bottom right corner shows correlations between nutrient variables. Unshaded top right corner shows correlations between life history and nutrient variables. The variables 'min depth' and 'max depth' refer to habitat preferences. 'Max length' refers to maximum reported length of the species. The variables 'a_lw' and 'b_lw' refer to the scalar and exponential parameters in the length-weight relationship equation $W = aL^b$, wherein W is weight and L is length. 'Trophic level' refers to the weighted mean of the trophic level of the diet of the species. See Table 1 for more details

differences are lower in the phylogenetic signal-only method (Table 4). Moreover, the phylogenetic prediction method can be readily applied to any species whose phylogenetic relationship to measured species has been estimated. Regression-based prediction requires this information in addition to information on species life history, although imputation methods may help to predict missing observations from a combination of other life history predictors and phylogenetic information[19].

Because our phylogenetic method performed well in validation tests, we used it to estimate missing nutrient values for all commercially valuable species in our original 371 species, as well as for all 7500+ unmeasured species in the *Actinopterygii* phylogeny (see Supplementary Data Files 2–10)[14]. Moreover, we used this approach to identify ray-finned fish families representing promising sources of key nutrients (Table 5). Some of the families we identify contain no or relatively few measured species, but our phylogenetic method predicts their potential to contain species rich in specific nutrients.

## Discussion

At least one-tenth of the global population is vulnerable to future deficiencies in micronutrient intake linked to the degradation of marine ecosystems[7]. Unsustainable fishing practices, the effects of climate change on sea temperature and dissolved oxygen content, and pollution are all major threats to a wide range of commonly consumed fish stocks. Quantifying the exact magnitude of these risks depends on knowing the nutrient content of key species—information that is extremely expensive to obtain, given the cost of directly evaluating nutrient composition data in a laboratory setting. In this study, we develop a modeling framework for using phylogenetic and life history information to predict nutrient content in fish species of the economically and nutritionally important class *Actinopterygii* (ray-finned fishes). We focus on nine nutrients—protein, total fat, omega-3 and omega-6 fatty acids, iron, zinc, vitamin A, vitamin D, and vitamin B12. Fish can be important sources of each of these, especially among coastal populations in the developing world.

We find that most nutrients exhibit substantial phylogenetic signal—i.e., covariance among species in nutrient values is proportional to their shared evolutionary history—and a model based solely on phylogenetic relationships and empirical estimates of phylogenetic signal provide reasonable predictions of species nutrient content. This phylogenetic signal-based model provides better predictive ability than the multiple regression

| | **(1)** | **(2)** | **(3)** | **(4)** | **(5)** | **(6)** | **(7)** | **(8)** | **(9)** |
|---|---|---|---|---|---|---|---|---|---|
| | **Protein** | **Total fat** | **Omega-3** | **Omega-6** | **Iron** | **Zinc** | **Vitamin A** | **Vitamin B12** | **Vitamin D** |
| Intercept | 14.27*** | 1.98*** | 0.70 | 0.21 | 0.90*** | 0.71*** | 4.11*** | 2.60*** | 2.48*** |
| | (11.06) | (4.22) | (1.80) | (1.83) | (4.67) | (3.92) | (3.47) | (4.81) | (3.59) |
| Max length | 0.0069* | −0.0003 | −0.0006 | −0.0001 | 0.0002 | −0.0006 | 0.0016 | −0.0004 | 0.0018 |
| | (2.52) | (−0.36) | (−1.04) | (−0.76) | (0.63) | (−1.82) | (0.90) | (−0.52) | (1.79) |
| Trophic level | 1.29*** | −0.17 | −0.0262 | 0.0196 | −0.07 | 0.01 | −0.43 | −0.24 | −0.21 |
| | (3.44) | (−1.54) | (−0.25) | (−0.62) | (−1.21) | (−0.16) | (−1.39) | (−1.59) | (−1.19) |
| Max depth | −0.0014*** | 0.0004*** | 0.0002** | 0.0001** | −0.0001 | −0.0001 | 0.0011*** | −0.0001 | 0.0001 |
| | (−3.99) | (4.30) | (3.21) | (2.87) | (−1.67) | (−1.30) | (5.22) | (−1.32) | (−0.61) |
| N | 183 | 183 | 81 | 81 | 175 | 111 | 101 | 89 | 93 |
| Pagel's λ | 0.000 | 0.627 | 0.672 | 0.637 | 0.165 | 0.607 | 0.725 | 0.480 | 0.784 |
| Log-likelihood | 34.99 | 53.89 | 15.31 | 9.83 | 9.34 | 19.82 | 33.82 | 25.86 | 32.29 |
| p-Value | <0.0001 | <0.0001 | 0.0018 | 0.0180 | 0.0219 | 0.0002 | <0.0001 | <0.0001 | <0.0001 |

**Table 2 Phylogenetic generalized least squares models predicting nutrient content**

T-statistics in parentheses. All outcome variables except protein are logged
***p < 0.001 **p < 0.01 *p < 0.05

**Table 3 Unconditional phylogenetic signal (Pagel's λ) of each nutrient across subsets of *Actinopterygii***

| Nutrient | N | λ | p (λ = 0, likelihood ratio test) |
|---|---|---|---|
| Protein | 270 | 0.4188 | <0.0001 |
| Total fat | 267 | 0.6627 | <0.0001 |
| Omega-3 fatty acids | 89 | 0.6036 | 0.0014 |
| Omega-6 fatty acids | 89 | 0.5366 | 0.1782 |
| Iron | 254 | 0.3461 | 0.0008 |
| Zinc | 146 | 0.8538 | <0.0001 |
| Vitamin A | 122 | 0.7444 | 1.0000 |
| Vitamin B12 | 102 | 0.5080 | 0.0008 |
| Vitamin D | 103 | 0.7640 | <0.0001 |

model (Table 4), although the latter accounts for both phylogenetic signal and life history trait values. This seemingly counterintuitive result can be explained by two factors. First, the sample size for estimating parameters of phylogenetic signal-based model is greater than that of phylogenetic regression model for all nutrients (see Tables 2, 3). Some of the increased predictive capacity of the phylogenetic signal model may be a consequence of better parameter estimation associated with a larger sample of species. Second, even though some life history variables (such as maximum depth) are significant predictors of some nutrients, the magnitude of effect is relatively small (Table 2). Therefore, inclusion of life history variables in nutrient predictions, as in the phylogenetic regression models, does not seem to offset the loss of sample size imposed by the additional requirement of having data on species life history. Note that we do not intend for our results to suggest that life history is unimportant in determining fish nutrient content, but rather that based on available data, incorporation of life history information does not improve predictions of species nutrient values. With the addition of more life history data and/or use of missing-data imputation techniques, phylogenetic regression models may ultimately yield better predictions than the phylogenetic signal model.

In addition, the method developed in this paper provides the basis for targeting new potential sources of key nutrients. The tendency for variation in nutrients to be phylogenetically structured means that groups of species that are closely related to nutrient-rich species are also reasonably likely to be nutrient-rich. As noted above, we explored this possibility by looking across predicted and observed nutrient values for more than 7000 ray-finned fish species, and identified families that are likely to contain species rich in key nutrients. The nutritional quality of the

members of some of these families is well known; scombrids, carangids, salmonids, and clupeids make large contributions to existing fisheries. However, we also identify as potentially nutrient-rich several fish families for which species nutrient content of their species is mostly unknown (Table 5). For example, our predictions suggest that exocoetids (flying fish), sphyraenids (barracuda), and centropomids (snook) may be high in protein even though protein content has been measured for no more than one species in any of these families. In addition, even though no species of muraenid, ophichthid, congrid, or channichthyid has been assayed for total fat content, their close phylogenetic relationships to species high in fats lead us to suggest that they may be good sources of this nutrient. To be clear, we recommend that our predictions be used to inform the selection of potentially nutrient-rich species for nutrient content measurement, and we caution against use of our predictions as strong statements on which species should be exploited for nutrition.

Moreover, our identification of nutrient-rich families is limited in some important ways. First, we could only predict nutrient content in species represented in the Rabosky et al. phylogeny, and so some families may be over- or under-represented in our predictions[14]. For example, some nutrient-rich families may not appear in Table 5 if they have relatively few species in the phylogeny. Conversely, some families may appear to contain a large number of nutrient-rich species because they are well sampled in the phylogeny and because they contain one or more species known to be high in particular nutrients or are closely related to species of high nutritional quality. In fact, our phylogenetic prediction method is unable to make distinctions between unmeasured species of the same phylogenetic distance from measured relatives, though new nutritional data would likely help resolve this issue. Finally, we acknowledge that our predictions are contingent on Rabosky et al.'s estimation of phylogenetic relationships among these ray-finned fishes[14], whose relation to the true phylogeny is unknown. Future work will refine these predictions as increasing phylogenetic information becomes available. Despite these limitations, we view our approach as a first step toward more precise estimates of the consequences of marine ecosystem transformation.

We identify several priorities in taking the conclusions of this research forward. First, improving the phylogenies of other classes important for human nutrition, including other marine seafood species as well as freshwater fish, is essential. Second, increasing the size of the nutrient validation sample—that is,

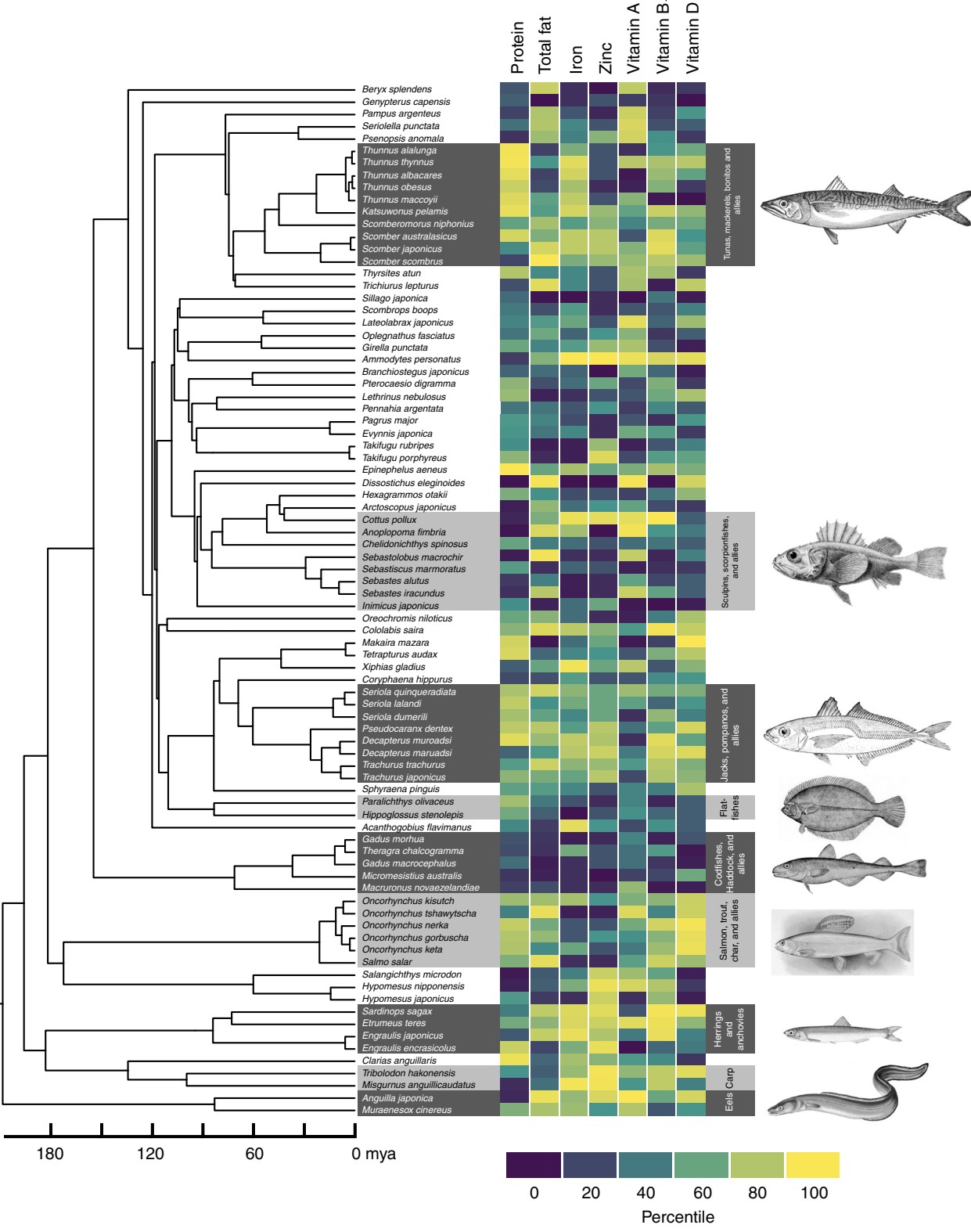

**Fig. 2** Phylogenetic relationships and nutrient content, expressed in percentiles. Eighty-four of the most commercially and nutritionally important of the 371 species in the database described above are shown. The time-scale is shown as millions of years ago (mya). The color scales indicate percentiles of nutrient content; see Supplementary Table 1 for ranges of each variable. Common names and pictorial representations of select groups of fishes are shown. All images are taken from Wikipedia, and are Public Domain images

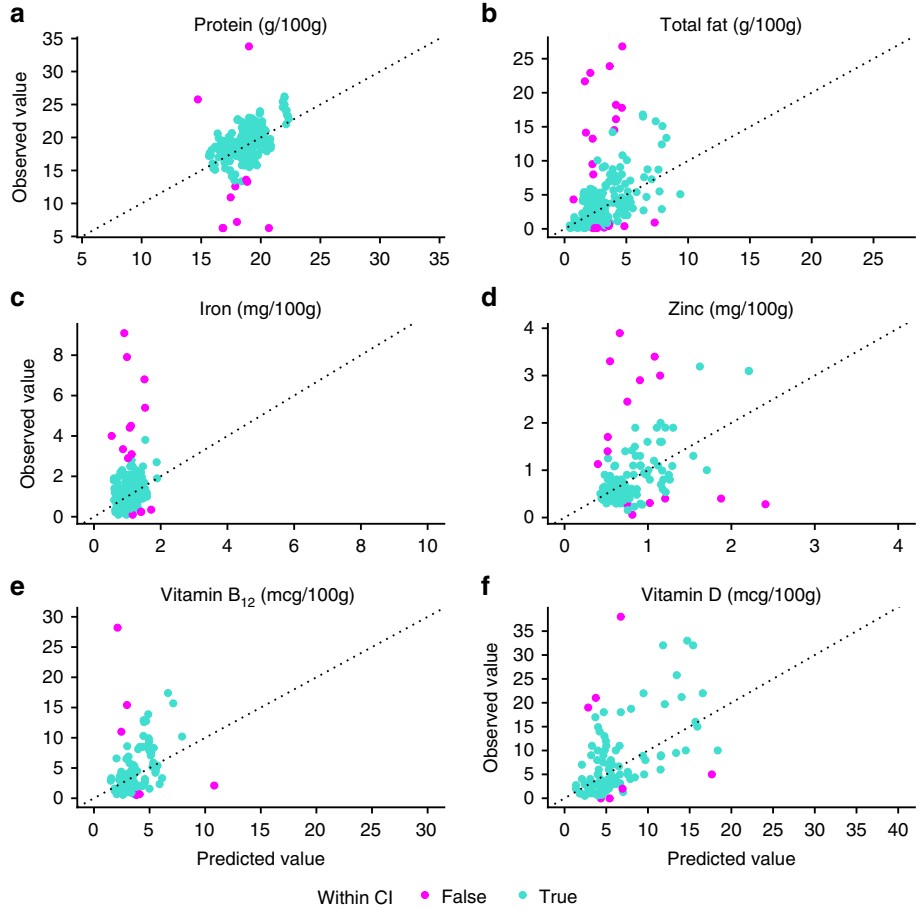

**Fig. 3** Predicted versus observed values for selected nutrients using phylogenetic signal only. Points in green fall within the 95% prediction interval; points in red are outside the interval. The diagonal represents points where predictions are equal to observed values

**Table 4 Confidence interval coverage and median deviation of nutrient prediction using phylogenetic signal only and phylogenetic signal plus life history information**

| | Phylogenetic signal only | | Phylogenetic signal + life history variables | |
|---|---|---|---|---|
| | 95% confidence interval coverage | Median difference (proportion of SD) | 95% confidence interval coverage | Median difference (proportion of SD) |
| Protein | 95.9 | 0.366 | 95.1 | 0.448 |
| Total fat | 92.1 | 0.304 | 91.8 | 0.316 |
| Iron | 94.5 | 0.395 | 97.1 | 0.473 |
| Zinc | 89.7 | 0.289 | 91.9 | 0.310 |
| Vitamin A | 91.0 | 0.036 | 88.1 | 0.034 |
| Vitamin B12 | 94.1 | 0.364 | 92.1 | 0.338 |
| Vitamin D | 93.2 | 0.346 | 91.4 | 0.347 |
| Mean | 92.9 | 0.300 | 92.5 | 0.323 |

Confidence interval coverage is the proportion of measured values falling inside the 95% prediction interval. Median percent difference is calculated as (measured value − predicted value)/sample standard deviation of species values

nutrient assays in species that belong to families identified as potentially nutrient-rich, as presented in Table 5. Third—approaching the small sample problem from the opposite direction—use of techniques like that of Thorson et al. (2017) to predict life history parameters would allow more existing nutrient information to be utilized[19].

We utilize several methods in this paper: evolutionary correlations, Brownian-motion models of trait evolution, and phylogenetic least squares regression. We believe that the availability of data is the key determinant of which of these methods is preferable in predicting fish nutrient content. In the absence of life history information, evolutionary correlations and phylogenetic signal-based models provide a rough sense of which nutrients are likely to be found together in *Actinopterygii*. As phylogenies expand and nutrient and life history datasets grow, life history regression models will likely improve on the predictive power of these methods. Overall, we believe that the combination of phylogenetic and life history information holds great potential for inexpensively inferring the nutrient content of a wide range of wild foods, and thereby quantifying the impacts of ecosystem transformation on human food and nutrition security.

## Methods

**Data sources**. We utilized fish nutrient information from food composition tables from Argentina, Bangladesh, Cambodia, Canada, Chile, Denmark, Japan, Finland, Gambia, Greece, Italy, Japan, Malaysia, South Korea, Peru, Turkey, and the United States; 89% of the species we examined have entries in either the Korea, Japan, Bangladesh, or Argentina tables[20–28,35–38]. The life history data come from

measuring the nutrient content of more species directly—would enable a better utilization of the large amount of life history information available. It is possible, perhaps even likely, that the weak associations between life history variables and nutrient content in this study is due to an overly restrictive validation sample. We recommend expanding the sample by prioritizing

**Table 5 Ray-finned fish families identified as potentially nutrient-rich based on phylogenetic prediction and observation of nutrient content**

| Nutrient | Family name | Types of fishes | Number of total species in family | Number of measured species in family | Number of nutrient-rich measured species in family |
|---|---|---|---|---|---|
| Protein | Serranidae | Sea basses | 111 | 7 | 2 |
| | Carangidae | Jacks, pompanos | 80 | 12 | 10 |
| | Scombridae | Tunas, mackerels | 37 | 22 | 18 |
| | Exocoetidae | Flying fishes | 32 | 1 | 1 |
| | Sphyraenidae | Barracudas | 14 | 1 | 1 |
| | Centropomidae | Snooks | 12 | 0 | 0 |
| | Salmonidae | Salmon, trout | 12 | 12 | 7 |
| | Istiophoridae | Billfishes | 8 | 1 | 1 |
| Total fat | Muraenidae | Moray eels | 56 | 0 | 0 |
| | Clupeidae | Herrings | 49 | 10 | 8 |
| | Nototheniidae | Cod icefishes | 28 | 1 | 1 |
| | Engraulidae | Anchovies | 22 | 6 | 4 |
| | Ophichthidae | Snake eels | 20 | 0 | 0 |
| | Scombridae | Tunas, mackerels | 19 | 21 | 10 |
| | Anguillidae | Freshwater eels | 15 | 1 | 1 |
| | Carangidae | Jacks, pompano | 14 | 12 | 6 |
| | Channichthyidae | Crocodile icefishes | 13 | 0 | 0 |
| | Congridae | Garden eels | 13 | 0 | 0 |
| | Salmonidae | Salmon, trout | 12 | 12 | 5 |
| | Centrolophidae | Medusafishes | 10 | 2 | 2 |
| Iron | Cyprinidae | Minnows, carp | 118 | 20 | 9 |
| | Cobitidae | Loaches | 102 | 2 | 2 |
| | Nemacheilidae | Stone loaches | 36 | 0 | 0 |
| | Balitoridae | River loaches | 34 | 0 | 0 |
| | Catostomidae | Suckers | 10 | 0 | 0 |
| | Scombridae | Tunas, mackerels | 10 | 19 | 10 |
| Zinc | Cyprinidae | Minnows, carp | 162 | 14 | 6 |
| | Cobitidae | Loaches | 102 | 1 | 1 |
| | Nemacheilidae | Stone loaches | 36 | 0 | 0 |
| | Balitoridae | River loaches | 34 | 0 | 0 |
| Vitamin A | Muraenidae | Moray eels | 56 | 0 | 0 |
| | Cyprinidae | Minnows, carp | 44 | 6 | 1 |
| | Cottidae | Sculpins | 32 | 1 | 1 |
| | Nototheniidae | Cod icefishes | 28 | 1 | 1 |
| | Mastacemblidae | Spiny eels | 21 | 1 | 1 |
| | Ophtichthidae | Snake eels | 20 | 0 | 0 |
| | Anguillidae | Freshwater eels | 15 | 1 | 1 |
| | Abyssocottidae | Deep water sculpins | 14 | 0 | 0 |
| | Channidae | Snakeheads | 14 | 1 | 1 |
| | Channichthyidae | Crocodile icefishes | 13 | 0 | 0 |
| | Congridae | Garden eels | 13 | 0 | 0 |
| | Centrolophidae | Medusafishes | 10 | 2 | 2 |
| Vitamin B12 | Cyprinidae | Minnows, carp | 136 | 4 | 2 |
| | Cobitidae | Loaches | 57 | 1 | 1 |
| | Clupeidae | Herrings | 54 | 4 | 4 |
| | Carangidae | Jacks, pompanos | 25 | 8 | 5 |
| | Engraulidae | Anchovies | 23 | 2 | 1 |
| | Salmonidae | Salmon, trout | 21 | 6 | 4 |
| | Osmeridae | Smelts | 13 | 2 | 2 |
| Vitamin D | Carangidae | Jacks, pompanos | 60 | 8 | 5 |
| | Muraenidae | Moray eels | 56 | 0 | 0 |
| | Clupeidae | Herrings | 54 | 6 | 6 |
| | Salmonidae | Salmon, trout | 30 | 8 | 7 |
| | Nototheniidae | Cod icefishes | 28 | 1 | 1 |
| | Cyprinidae | Minnows, carp | 20 | 3 | 1 |
| | Anguillidae | Freshwater eels | 15 | 1 | 1 |
| | Channichthyidae | Crocodile icefishes | 13 | 0 | 0 |
| | Sphyraenidae | Barracudas | 11 | 1 | 1 |

Final column lists number of species in each family identified as potentially nutrient-rich, based on phylogenetic prediction and observation of nutrient content. Species are considered nutrient-rich if their nutrient content rank within the 500 largest values out of more than 7000 species from the Rabosky et al. phylogeny with predicted or known nutrient content[14]. Values in parentheses indicate the number of measured species that are nutrient-rich over the total number of species measured for that family

FishBase, a publicly accessible database containing taxonomic, biological, ecological, life history, and human use information on finfishes[39]. We utilized the species-level, time-calibrated, multi-gene phylogeny for *Actinopterygii* assembled by Rabosky et al.[14]. Although other large-scale species-level phylogenetic trees are also available for this group[40,41], we chose the Rabosky et al. phylogeny because it maximizes overlap with nutrient content data. Rabosky et al.'s reconstruction of the phylogeny was based on maximum likelihood phylogenetic analysis of 13 genes with subsequent smoothing of among-lineage substitution rate heterogeneity and divergence date estimation using fossil calibrations. The original phylogeny includes over 7500 ray-finned fish species. As described in more detail further below, we used this phylogeny in combination with life history predictor variables and estimates of phylogenetic signal (i.e., the tendency for phenotypic similarity among species to be proportional to their time of shared evolution) to predict nutrient content in species lacking such information.

**Evolutionary correlations**. We first explored bivariate evolutionary correlations between all pairwise combinations of life history and nutrient variables. Here, evolutionary correlations are the Pearson correlation coefficients describing associations between evolutionary changes in pairs of variables[42,43]. We estimated these correlations for log-transformed species values given the Rabosky et al. phylogeny and a Brownian motion model with Pagel's $\lambda$ correction for the degree of phylogenetic signal using the *ratematrix* function of the *geiger* package[44] for R[45].

**Multivariable regression models**. To evaluate the capacity to predict nutrient content from life history information, we fit multiple regression models using phylogenetic least squares (PGLS) as implemented in the R package *phylolm*[46]. Because species with a recent common ancestor are expected to have more similar trait values, the assumptions of ordinary least squares (OLS) are violated[47]. PGLS accounts for phylogenetic non-independence using shared ancestry as inverse weights on the elements of the residual variance-covariance matrix used in the model[15,29–31]. Thus, in matrix notation, a coefficient $\beta$ is estimated as follows:

$$\beta = \left(\mathbf{X}' V^{-1} \mathbf{X}\right)^{-1} \mathbf{X}' V^{-1} \mathbf{y} \qquad (1)$$

where X is a matrix of *n* species and *m+1* life history trait values (with an intercept column); X′ is the transpose of X; *y* is a vector of values for a given nutrient; and V is the residual variance-covariance matrix. Under OLS assumptions, the diagonal elements of V are the variance of the residuals; the residuals are expected to be normally distributed with mean zero. Under PGLS, the residual covariances are computed using branch lengths from each member of a species pair to their common ancestor. Instead of assuming covariance among species to be proportional to their time of shared evolution, we used a maximum likelihood procedure to identify a scalar of $V^{-1}$, called Pagel's $\lambda$, that best fits the observed data; $\lambda = 0$, the OLS approximation, would indicate trait evolution completely independent of phylogeny, $\lambda = 1$ would indicate that shared evolutionary history (i.e., shared phylogenetic branch length) predicts phenotypic similarity among species, and intermediate $\lambda$ ($0 < \lambda < 1$) discounts the phylogenetic dependence of trait values among species[30,32,33]. The best-fit inverse matrix $V^{-1}$ is then used to estimate the predictor coefficient $\beta$[30,31,48].

**Phylogenetic prediction**. The objective of the PGLS modeling exercise is to advance towards methods for predicting the nutrient content of species for which information is not available. Although our analysis shows that life history parameters are generally weak predictors of all nutrients, phylogeny is more promising. We thus predict nutrient content of unmeasured *Actinopterygii* fish species using nutrient data for measured species and the Rabosky et al. phylogenetic tree relating both measured and unmeasured species[14], assuming a $\lambda$-corrected Brownian motion model of evolution. Under this model, character change along any branch of the phylogeny is a normally distributed random variate with expected value equal to zero and variance proportional to the length of the branch[24]. Because the character has no tendency to increase or decrease under the model, the predicted value for an unmeasured species is equal to the estimated state for the most recent common ancestor (MRCA) between that species and the measured species most closely related to it. The state of this ancestor is evaluated as the branch length-weighted mean of the estimated character states in the next node deeper and the next more recent node that connect measured species (though, in some cases, the more recent node is a tip species). We estimated the states of these nodes using the *phytools* function *fastAnc*[49].

Although the expectation under the Brownian model is that the character will remain unchanged in the unmeasured species from the time of its split from the measured species most closely related to it, the uncertainty in the predicted value increases with time since this split. Therefore, the variance around each predicted value incorporates the branch length (*t*) subtending the unmeasured species, and is evaluated as

$$t \times \sigma^2 \qquad (2)$$

where $\sigma^2$ is the time-independent variance of the Brownian motion process[24]. The parameter $\sigma^2$ was estimated from the data and phylogeny using methods described

in the next paragraph. We constructed 95% confidence intervals around each predicted value as $\pm 1.96 \times \sqrt{t \times \sigma^2}$.

Our predictions also incorporate empirical estimates of the degree of phylogenetic signal in measured species nutrient values. We fit Pagel's $\lambda$ separately for each nutrient using the *phylosig* function of the *phytools* package for R[22,32,33,44], which returns the maximum likelihood estimates for both $\sigma^2$ and $\lambda$. We then transformed branch lengths of the phylogeny by the empirical estimates for $\lambda$ using the *rescale* function of the R package *geiger*;[21] $\lambda = 1$ recovers the original phylogeny, $\lambda < 1$ compresses internal branches relative to terminals, and $\lambda = 0$ is a star phylogeny. We then estimated ancestral character states, predicted states for unmeasured species, and their confidence intervals on this branch length-transformed tree.

**Validation**. We used a jackknifing approach to validate the method for phylogenetic prediction of nutrient content. For each nutrient, we removed one measured species from the dataset and applied the method to predict the species' nutrient value and calculate its prediction interval. We then determined whether the prediction interval contained the measured value. If it does not, we label that trial as an error, and then calculated the error rate for each nutrient over all measured species. We also calculated the median percent deviation between predicted and measured values as a proportion of the standard deviation of the sample of all species nutrient values. We then compared error rates and accuracy of phylogeny-only predictions to predictions based on the best-fit multiple regression models. These latter predictions were obtained following the method of Garland and Ives[50], and were restricted to the sample of species for which life history data were available.

**Code availability**. All code used in the analysis is made available as Supplementary Data 11–15. SD11 is the script for evolutionary correlations. SD12 is the script for phylogenetic least squares. SD13 is the script for estimating the phylogenetic signal of nutrient variables. SD14 is the script for validating the predications under the lambda model. SD15 is the script for validating predictions under the lambda plus phylogenetic regression model.

## Data availability

We declare all data used in the above analysis, tables, and figures to be available within the paper and in the supplementary information files. Supplementary Data 1 is the fish life history and nutrient content database. Phylogenetically predicted nutrient values are labeled as Supplementary Data 2–10.

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

## Acknowledgements

We would like to thank the Wellcome Trust for their financial support through the Our Planet, Our Health program [Grant number: 106864MA].

## Author contributions

B.V. led writing of the manuscript and contributed to the analysis. D.C. led analysis. M.R. S. and S.S.M. created the underlying fish nutrient database. B.L.R. contributed to draft revisions and designed Fig. 2. C.D.G. conceptualized the research idea and overall study design, and all authors contributed to writing and draft revision.

## Additional information

**Competing interests:** The authors declare no competing interests.

