## [Peer Review File · Nature Communications]

Reviewers' comments:

Reviewer #1 (Remarks to the Author):

Overall this is an interesting piece of work that could make an important contribution to the literature; however, there are aspects of the work that need to be clarified prior to the manuscript being acceptable for publication. I have provided overall and specific comments below primarily as it relates to the nutrition components of this study.

Overall comments:

- When the authors refer to lipids, it's not clear which lipids they are talking about. In general, fish tends to be a rich source of unsaturated fatty acids. If the data allows for it, it would be helpful to look at the different types of fatty acids separately rather than lumping them altogether. In particular, it would be helpful to look at essential fatty acids (omega 3 and 6s) which are high in some fish species and have important health benefits.
- I would argue that the authors are looking at food and nutrition security rather than simply food security on its own. I think the manuscript should be amended to say 'food and nutrition security' throughout.

Specific Comments:

Abstract

Line 15: The authors should say "potential changes in human nutrient...". I think it's necessary to be a bit more conservative with the wording based on the study's actual findings. I also think it would be helpful for the authors to take the conclusions of the abstract a bit further (and those of the paper as well). What is the next step in terms of taking the findings of this paper forward in a practical and meaningful way? I understand that there may not be room in the abstract but in the body of the paper this should be emphasized more.

Introduction

Lines 31-33: The authors need to be careful with the wording in this sentence. Deficiencies are not leading to increased risk of CHD. Fish consumption can be cardio-protective, particularly for secondary prevention, but this is not based on a deficiency. I think the wording should be changed to reflect the two main benefits of fish consumption: 1) ensuring that populations get important macro and micronutrients (particularly in the context of undernutrition) and 2) the potential overall health benefits of consuming fish in the context of non-communicable disease (NCDs) and health promotion. The latter point raises another issue in terms of sustainability. If people were to consume the amount of fish that has been suggested in order to prevent NCDs, it would put a huge strain on the fish supply and would lead to other challenges in terms of sustainability. There is a tradeoff between health and sustainability when it comes to fish.

Line 35: What information is available regarding the nutrient content of fish?

Line 41: How was it determined that ray-finned fishes are the most nutritionally important? It would be helpful to provide a reference.

Results

Line 55-56: "Each of these nutrients is present in relatively high concentrations". I think this should be reworded to make it clear that although many of these nutrients are present in high concentrations they aren't all, as the authors' analyses have demonstrated.

Table 1: I think the authors should consider using the Dietary Reference Intake publications to describe the different macro and micronutrients. These contain comprehensive descriptions of the role of all the macro and micronutrients. It would greatly strengthen the table. For example, I don't think that the description for lipids is strong and neglects key aspects of lipids. As I mentioned in the overall

comments, I think there needs to be some recognition of the different types of fatty acids.

Line 66: How was "valid information" determined?

Lines 66-70: Additional information is required about how "sources" and "rich sources" were determined. I understand from Supplementary Table 3 that CODEX thresholds were used but what are the CODEX thresholds based on? Is it a certain percentage of daily value or of the RDA? This information should be provided somewhere in the manuscript or in the supplementary files.

Lines 114-116: Although the authors are correct that in many cases only a small number of nutrients are included in food composition databases this is not always the case. The INFOODS food composition table for fish has several nutrients: <http://www.fao.org/infoods/infoods/tables-and-databases/faoinfoods-databases/en/>

Line 121-22: Is there a reference for the statement that smaller fish are often overlooked? It may be worth mentioned the push by some organizations, including worldfish, to highlight the importance of small fish species for nutrition, particularly among vulnerable populations including women and young children.

Table 5: What was the threshold for determining nutrient-rich based on? Is that an accepted way of defining nutrient-rich?

Discussion

Line 213-14: As mentioned previously, it is not entirely accurate to state that it will lead to fatty acid deficiencies from a nutrition perspective. I realize that the reference that was cited does use similar wording but it would be more accurate to refer to essential fatty acids (rather than fatty acids overall) explicitly. It is actually quite rare to have a deficiency in essential fatty acids but it more likely that consumption levels are inadequate in terms of prevention CHD, etc. It may be worth clarifying that.

Lines 222-23: I think this should be re-worded to "fish can be important.."

Lines 233-34: Can the authors comment on the usefulness of these predictions, given the text in lines 250-52. What exactly are the next steps stemming from this research?

Methods

Lines 274-77: Why were these food composition tables used? Why not use the INFOODS food composition tables for fish and seafood? <http://www.fao.org/infoods/infoods/tables-and-databases/faoinfoods-databases/en/>

Reviewer #2 (Remarks to the Author):

Major comments (from James Thorson, declining anonymity):

In this paper, the authors use a publicly available database of nutrient-content data for fishes obtained from the Food and Agricultural Organization (FAO), analyze covariation between nutrient contents, life-history parameters, and taxonomy, and predict nutrient contents for a large portion of bony fishes worldwide. I agree with the authors that results from analysis could be useful in future analyses regarding the likely impact of global changes on nutrient availability.

However, I note several ways in which the analysis could be improved via changes in analytical methods.

1. The authors don't appear to use multivariate methods for their phylogenetic analysis or predictions. I.e., Eq. 2 describes a random-walk model for trait-evolution for each nutrient individually. Using a multivariate model would be important to improve confidence-interval coverage, given that errors will significantly covary given the observed correlations (Fig. 1), and this is important to the paper given the authors' use of confidence-interval coverage as a performance metric (in Table 4). A multivariate model would also be useful for future users, so that predictive intervals capture covariance among traits. Finally, a multivariate model could be used to illuminate major axes of trait evolution. For example, a multivariate model could be combined with major axis regression to identify a "dominant

axis" of trait evolution (Warton, Wright, Falster, & Westoby, 2006; Thorson, Munch, Cope, & Gao, 2017). Given the correlation matrix (Fig. 1), I imagine the major nutrient trade-off is between protein and fats.

2. The authors could improve their model validation techniques. Although I appreciate the use of interval-coverage and median relative error, most statisticians would prefer some use of "predictive scores" to evaluate model fit, as well as plots of model residuals. In a multivariate model, residuals can be shown via bivariate predictive ellipses, or via a Chi-squared statistic for each vector of residuals.

3. The authors appear to use a variety of different methods including descriptive (sample correlation, Fig. 1), model-based (random-walk model for trait evolution, Fig. 1), and regression (phylogenetic-corrected least squares). However, there is relatively little comparison among methods, or illustration of how these methods could be combined in future studies. One option which would likely be useful is a phylogenetic method that includes missing values, where the vector of life-history parameters and nutrient contents would be analyzed jointly. Major axis regression would then show the "slope" between nutrients and life-history parameters, and nutrients could still be predicted for species with unknown phylogeny (i.e., based on the measured life-history traits for that species). Alternatively, major axis regression could be used for both the correlation matrix (Fig. 1), and the multivariate phylogenetic analysis (an extension of Eq. 1), and estimated slopes could be compared with regression results from the phylogenetic-corrected least squares method.

Minor comments:

21 – My memory of references 2-3 is that they are forecasts of potential future changes. I believe there are other papers that are specifically focused on documenting past changes (Perry, Low, Ellis, & Reynolds, 2005; Pinsky, Worm, Fogarty, Sarmiento, & Levin, 2013, etc.)

25 – I think this paper documents "impacts in every marine ecosystem" but not "degradation".

78-79 – I don't see any evidence that the variance in your data set is representative of fishes in general.

107-110 – Again, these claims would be easier to quantify using major axis regression with the sample covariance. It appears that there is a trade-off between fat and protein. Using log-scaled variables, the ratio X of eigen-vector elements can be interpreted as "a 1% change in variable A is associated with a $X\%$ change in variable B".

128-131 – these would be interesting to compare with major axis regression results.

135 – Please add "... when phylogenetic information are available."

176-177 – Please delete sentence and reference figure when explaining specific results.

226 – Please add "... when comparing results between a Brownian-motion model for trait evolution and a phylogenetic least squares approach". I'm not convinced that you have explored the full suite of possible models, so an absolute statement of relative information between life-history and phylogeny is not supported.

261 – This concern is exactly why it is beneficial to use a phylogenetic model that can handle missing values on both traits and nutrient-content samples. In this case, two unmeasured species with equivalent distance from the nearest measured relatives would still have their predictions updated by known life-history traits.

315 – I think you mean the residuals are expected to be normally distributed.

353 – What about covariance? There is clearly covariance among traits (Fig. 1). This can be informative and interpretable.

Works cited

Perry, A. L., Low, P. J., Ellis, J. R., & Reynolds, J. D. (2005). Climate Change and Distribution Shifts in Marine Fishes. *Science*, 308(5730), 1912–1915. doi:10.1126/science.1111322

Pinsky, M. L., Worm, B., Fogarty, M. J., Sarmiento, J. L., & Levin, S. A. (2013). Marine taxa track local climate velocities. *Science*, 341(6151), 1239–1242.

Thorson, J. T., Munch, S. B., Cope, J. M., & Gao, J. (2017). Predicting life history parameters for all fishes worldwide. *Ecological Applications*, 27(8), 2262–2276. doi:10.1002/eap.1606

Warton, D. I., Wright, I. J., Falster, D. S., & Westoby, M. (2006). Bivariate line-fitting methods for allometry. *Biological Reviews*, 81(2), 259–291. doi:10.1017/S1464793106007007

Reviewer #3 (Remarks to the Author):

This study examines the evolution of nutrients relevant to human health found in major fish lineages of commercial importance. The study relies upon a previously published phylogeny of fishes (Rabosky et al., 2013) and literature sources of nutrient information for the comparative analyses and builds a novel predictive model of nutrient content for 7500+ ray finned fishes. I found the application of PCMs to nutrients and human health novel and fascinating. Although I think the study is missing some details about the analyses I think that in general the results are interesting, novel, and deserve publication. However I am not yet convinced that the story as presented here translates to a wide audience instead of, say, a journal more focused on fisheries. I agree that the phylogenetic associations reported for the evolution of the nutrients themselves are novel and that the tie in to human health and nutrition is potentially far-reaching. However right now I am missing justification, details, and possibly analyses that make a stronger case for publication in this journal. Here is a list of major and minor concerns...

- in Supplementary Tables 2 and 3 what is the link between these tables and the importance of fish to low income coastal populations?
- is the list of species below SI Table 3 part of Table 3? Why does this table contain non-actinoptes (Meretrix, Ruditapes)?
- what is known about the correlation of these nutrients in non-phylogenetically controlled studies? For example, the paper reports that 98% of species are rich sources of protein (and it looks like all are sources of protein). Under what conditions could a fish species not be a good source of protein? Are vitamins A and D known to co-occur in commonly fished species already? Is this association with fat noted as well?
- are there any physiological, life history explanations for the evolutionary trends you report (lines 10-110)? Why can vitamin A concentration can evolve so quickly? Why are other traits slow to evolve?
- Figure 2 needs to be made much more accessible to a general readership. Designating the higher level taxonomy with common names of the groups (Scombriforms-tunas, mackerels, and allies as an example) would help readers understand the phylogenetic scope of the analysis. There should be a true timescale on the x axis. Representative illustrations of the major groups would also be helpful. These kinds of changes would help avoid the way the text currently directs the reader to results on the tree (lines 151-159 for example).
- Figure 3 yes your model allows you to reasonably predict nutrient content. However can you comment on the biological implications of width of the confidence envelope. For example in SI Table *Sebastes_taczanowskii* is within the model confidence limit for lipids but the predicted and measured values differ by an order of magnitude. Does this limit the utility of the phylogenetic prediction model?
- Figure 3 can you say anything about the species that fall outside the model limits? Also, can you comment on why the model is consistently underestimating the nutrient content in its predictions?
- I could not find the data file for the 7500+ species that are predicted (apologies if it was included but I could not find it in the SI).
- l 48-49 citation for source(s) of data?
- l116 this strikes me as quite vague. If observed correlations do indeed predict unmeasured nutrients then this extends the reach of the paper. Are there any citations here that support your statements? Are there any subsets of the data in hand that could demonstrate the ability of your analyses to predict levels of unobserved nutrients?
- The authors should provide a data file with the raw values of the nutrient levels and a citation to the source of the data.

- I would like to see the SI include the tree file, data files, and scripts used to conduct the analyses so that they may be repeated.

Reviewers' comments:

Reviewer #1 (Remarks to the Author):

Overall this is an interesting piece of work that could make an important contribution to the literature; however, there are aspects of the work that need to be clarified prior to the manuscript being acceptable for publication. I have provided overall and specific comments below primarily as it relates to the nutrition components of this study.

Overall comments:

- When the authors refer to lipids, it's not clear which lipids they are talking about. In general, fish tends to be a rich source of unsaturated fatty acids. If the data allows for it, it would be helpful to look at the different types of fatty acids separately rather than lumping them altogether. In particular, it would be helpful to look at essential fatty acids (omega 3 and 6s) which are high in some fish species and have important health benefits.

Changed “lipids” to “total fat.” We agree with the review on the importance of fatty acids, and re-do all analyses not restricted by sample size for omega-3 and omega-6 fatty acids. We report omega-3 and omega-6 fatty acids phylogenetic least squares analysis in the main text. We also include omega-3 and omega-6 fatty acids in the evolutionary correlation matrix in Supplementary Table 5 (because of sample size restrictions, we exclude them in Figure 1’s correlation matrix of the main paper). We estimate the unconditional phylogenetic signal of fatty acids in Table 3. Because of sample size issues, and because omega-3 and omega-6 are very strongly correlated to total fat—and to each other—we do not carry out prediction validation or the full set of predictions for these variables.

- I would argue that the authors are looking at food and nutrition security rather than simply food security on its own. I think the manuscript should be amended to say ‘food and nutrition security’ throughout.

Changed all instances in text to “food and nutrition security.”

Specific Comments:

Abstract

Line 15: The authors should say “potential changes in human nutrient...”. I think it's necessary to be a bit more conservative with the wording based on the study's actual findings.

Changed to “potential changes...”

I also think it would be helpful for the authors to take the conclusions of the abstract a bit further (and those of the paper as well). What is the next step in terms of taking the findings of this

paper forward in a practical and meaningful way? I understand that there may not be room in the abstract but in the body of the paper this should be emphasized more.

Added final paragraph (just before methods section):

“We identify several key steps in taking the conclusions of this research forward. First, improving the phylogenies of other classes important for human nutrition, including other marine seafood species as well as freshwater fish, is essential. Second, increasing the size of the nutrient validation sample—that is, measuring the nutrient content of more species directly—would enable a better utilization of the large amount of life history information available. It is possible, perhaps even likely, that the weak associations between life history variables and nutrient content in this study is due to an overly restrictive validation sample. Third—and approaching the small sample problem from the opposite direction—use of techniques like that of Thorson et al. (2017) to predict life history parameters would allow more existing nutrient information to be utilized. Overall, we believe that the combination of phylogenetic and life history information holds great potential for inexpensively inferring the nutrient content of a wide range of wild foods, and thereby quantifying the impacts of ecosystem transformation on human food and nutritional security.”

Introduction

Lines 31-33: The authors need to be careful with the wording in this sentence. Deficiencies are not leading to increased risk of CHD. Fish consumption can be cardio-protective, particularly for secondary prevention, but this is not based on a deficiency. I think the wording should be changed to reflect the two main benefits of fish consumption: 1) ensuring that populations get important macro and micronutrients (particularly in the context of undernutrition) and 2) the potential overall health benefits of consuming fish in the context of non-communicable disease (NCDs) and health promotion. The latter point raises another issue in terms of sustainability. If people were to consume the amount of fish that has been suggested in order to prevent NCDs, it would put a huge strain on the fish supply and would lead to other challenges in terms of sustainability. There is a tradeoff between health and sustainability when it comes to fish.

Changed to reflect guidance above. Revised paragraph reads:

“The consequences of these changes for human health, and especially for nutritional intake, are likely to be severe because fish provide critical micronutrients essential to human nutrition, including iron, zinc, vitamin A, vitamin B12, certain fatty acids, and others.^{i,iii,iii} In many societies, seafood is the foundation for healthy diets, and its decline presents a significant risk in destabilizing food and nutrition security.^{iv} The consumption of fish results in a wide range of health benefits, including the prevention of various non-communicable diseases and the promotion of cognitive development.^{v,vi}”

Line 35: What information is available regarding the nutrient content of fish?

We collected all available Actinopterygii information in the database used for this paper, consolidating national-level nutrient composition databases, primary sources, and other

meta-databases. To our knowledge, our dataset represents the most comprehensive source available for Actinopterygii.

Line 41: How was it determined that ray-finned fishes are the most nutritionally important? It would be helpful to provide a reference.

We assume this based on total capture fisheries weight represented by the sample we use; we include the following line in the first paragraph of the results, with associated FAO reference:

“The set of 376 species represents over half of all global capture fisheries by weight, with all 22 of the world’s most harvested marine fish species included^{vii}.”

Results

Line 55-56: “Each of these nutrients is present in relatively high concentrations”. I think this should be reworded to make it clear that although many of these nutrients are present in high concentrations they aren’t all, as the authors’ analyses have demonstrated.

Changed in text. New sentence reads:

“These nutrients are generally present in relatively high concentrations in seafood—although, as we describe below, not all the above nutrients are present in appreciable quantities in all species—and are critical for human nutrition.”

Table 1: I think the authors should consider using the Dietary Reference Intake publications to describe the different macro and micronutrients. These contain comprehensive descriptions of the role of all the macro and micronutrients. It would greatly strengthen the table. For example, I don’t think that the description for lipids is strong and neglects key aspects of lipids. As I mentioned in the overall comments, I think there needs to be some recognition of the different types of fatty acids.

Changed. All nutrient descriptions taken from DRI/National Academies publications, and omega n-6/n-3 fatty acids added.

Line 66: How was “valid information” determined?

Changed to “available information,” which is the intended meaning. Information is determined valid if taken from a nutrient composition table generally considered of acceptable quality (i.e., originating from government or academic sources).

Lines 66-70: Additional information is required about how “sources” and “rich sources” were determined. I understand from Supplementary Table 3 that CODEX thresholds were used but what are the CODEX thresholds based on? Is it a certain percentage of daily value or of the

RDA? This information should be provided somewhere in the manuscript or in the supplementary files.

These are indeed definitions used by FAO in the construction of the Codex Alimentarius International Food Standards, but other than the general statement "Codex standards are based on sound science provided by independent international risk assessment bodies or ad-hoc consultations organized by FAO and WHO", we were not able to locate further information. We are contacting FAO about this.

Lines 114-116: Although the authors are correct that in many cases only a small number of nutrients are included in food composition databases this is not always the case. The INFOODS food composition table for fish has several nutrients: <http://www.fao.org/infoods/infoods/tables-and-databases/faoinfoods-databases/en/>

We only included INFOODS-related entries when we lacked data from other sources about a given species. Because uFish and AnFoodD (databases related to INFOODS) databases in the past were aggregated from data in research articles, they rarely collected a full suite of nutrients for a given fish — they only included data for whatever they happened to be studying. However, in the newer uFish database, the compilers aggregated various nutrient data points for each individual fish type from a variety of different sources (and thus specimens) under a single species entry. This is helpful in a general reference sense, but it is not very useful from our standpoint: we are trying to look at the associations between the various nutrient contents of individual fish, which can vary by season, location, growing conditions, etc. And in cases when the uFish entries are taken from several different fish, we were not confident that the associations between different nutrients, taken from different specimens, were correlated in the same way as for an individual specimen.

We also mistakenly excluded an INFOODS-related reference (Rittenschober et al 2013, below); now added.

Rittenschober D, V Nowak, and UR Charrondiere. 2013. Review of availability of food composition data for fish and shellfish. *Food Chemistry* 141, pp.4303-4310.

Line 121-22: Is there a reference for the statement that smaller fish are often overlooked? It may be worth mentioned the push by some organizations, including worldfish, to highlight the importance of small fish species for nutrition, particularly among vulnerable populations including women and young children.

Added reference (the researchers are affiliated with WorldFish):

Bogard, Jessica R., Shakuntala H. Thilsted, Geoffrey C. Marks, Md Abdul Wahab, Mostafa AR Hossain, Jette Jakobsen, and James Stangoulis. "Nutrient composition of important fish species in Bangladesh and potential contribution to recommended nutrient intakes." *Journal of Food Composition and Analysis* 42 (2015): 120-133.

We have included information from this paper where relevant; however, the researchers' focus is on inland fish and not Actinopterygii generally.

Table 5: What was the threshold for determining nutrient-rich based on? Is that an accepted way of defining nutrient-rich?

FAO Codex Alimentarius standards; please see response to comment on Lines 66-70 above.

Discussion

Line 213-14: As mentioned previously, it is not entirely accurate to state that it will lead to fatty acid deficiencies from a nutrition perspective. I realize that the reference that was cited does use similar wording but it would be more accurate to refer to essential fatty acids (rather than fatty acids overall) explicitly. It is actually quite rare to have a deficiency in essential fatty acids but it more likely that consumption levels are inadequate in terms of prevention CHD, etc. It may be worth clarifying that.

Eliminated fatty acid reference in this sentence. In initial section, reworded "deficiency" language to "The consumption of fish results in a wide range of health benefits, including the prevention of various non-communicable diseases and the promotion of cognitive development."

Lines 222-23: I think this should be re-worded to "fish can be important..."

Changed in text

Lines 233-34: Can the authors comment on the usefulness of these predictions, given the text in lines 250-52. What exactly are the next steps stemming from this research?

Please see response to comment on abstract comment above; added concluding paragraph.

Methods

Lines 274-77: Why were these food composition tables used? Why not use the INFOODS food composition tables for fish and seafood? <http://www.fao.org/infoods/infoods/tables-and-databases/faoinfoods-databases/en/>

Please see response to line 114-116 comment above.

Reviewer #2 (Remarks to the Author):

Major comments (from James Thorson, declining anonymity):

In this paper, the authors use a publicly available database of nutrient-content data for fishes obtained from the Food and Agricultural Organization (FAO), analyze covariation between nutrient contents, life-history parameters, and taxonomy, and predict nutrient contents for a large portion of bony fishes worldwide. I agree with the authors that results from analysis could be useful in future analyses regarding the likely impact of global changes on nutrient availability. However, I note several ways in which the analysis could be improved via changes in analytical methods.

1. The authors don't appear to use multivariate methods for their phylogenetic analysis or predictions. I.e., Eq. 2 describes a random-walk model for trait-evolution for each nutrient individually. Using a multivariate model would be important to improve confidence-interval coverage, given that errors will significantly covary given the observed correlations (Fig. 1), and this is important to the paper given the authors' use of confidence-interval coverage as a performance metric (in Table 4). A multivariate model would also be useful for future users, so that predictive intervals capture covariance among traits. Finally, a multivariate model could be used to illuminate major axes of trait evolution. For example, a multivariate model could be combined with major axis regression to identify a "dominant axis" of trait evolution (Warton, Wright, Falster, & Westoby, 2006; Thorson, Munch, Cope, & Gao, 2017). Given the correlation matrix (Fig. 1), I imagine the major nutrient trade-off is between protein and fats.

The reviewer presents an intriguing idea—one that we investigated in an early phase of our study. Given the observed correlations among nutrients, we were interested in the predictive capacity of phylogeny and life history variables for major axes of nutrient content variation and covariation. We performed phylogenetic principal components analysis (PPCA)—principal components analysis that incorporates phylogenetic relationships and an estimate of phylogenetic signal. This analysis led to two components with eigenvalues greater than 1:

	PC 1	PC 2	PC 3	PC 4
protein	0.68	0.27	0.42	0.03
total fat	-0.76	-0.18	0.03	0.47
iron	-0.25	0.76	-0.25	-0.40
vitamin A	-0.73	-0.19	0.16	-0.44
vitmin D	-0.32	0.41	0.79	0.05
vitamin B12	-0.18	0.78	-0.27	0.35
Eigenvalue	1.76	1.49	0.96	0.70
% Total Var	29.3	24.8	16.0	11.7

$\Lambda = 0.59 \rightarrow$ moderate level of phylogenetic signal

As the reviewer noted, PC1 was associated with a negative relationship between protein and fat (and between protein and other nutrients), with this axis separating species high in protein and low in other nutrients from those low in protein but high in other nutrients. PC2 was most strongly associated with higher iron and vitamin B12, though this axis is much more difficult to interpret functionally. We took PC1 to be a meaningful axis of

species nutrient content that captures prominent correlations, and analyzed PC1 in the same manner as the individual nutrients (fitting regression models with PGLS, predicting unmeasured species values, and carrying out prediction validation). However, analyses of this axis gave mixed results.

Analysis of PC1 revealed patterns similar to those of the individual variables on which it loads strongly (protein and fat, in particular), with moderate phylogenetic signal ($\lambda = 0.65$, $P(\lambda = 0) = 0.017$) and significant effect of maximum depth ($b1 = -0.011 \pm 0.003$, $t = 4.28$, $P < 0.001$) but non-significant effects of other life history variables. However, the prediction validation exercise revealed that predicted PC1 scores for species were less accurate (predictions deviated from measured values by about 0.5 sample standard deviations) and more prone to error (measured values were outside prediction intervals in about 10% of species). Because analyses involving PC1 do not shed additional light on our central questions, we have left it out of the manuscript.

In addition, we also explored the possibility that subgroups of ray-finned fish cluster differently along PCs, and we examined scatterplots of species' principal component scores. However, we did not observe a clear pattern of variation within or among major groups of ray-finned fish:

We would like to clarify one additional consideration relevant to this analysis: although we think this approach is potentially promising, this method suffers from significant data limitations. To be included in the PPCA, species must satisfy two criteria: the species (1) has been measured for all nutrient variables and (2) is included in the Rabosky et al. (2013) phylogeny. Our data set includes 85 species meeting these criteria. Moreover, fitting regression models with life history predictors further requires that life history data are also available. This additional requirement further reduces our data set to 78 species, which is less than the sample size available for most individual nutrients. We suspect that the weak results and predictions involving PCs is a consequence of low sample size due to data limitation—a point we now make explicitly in the manuscript, though in a somewhat different context (see response to comment 3 below).

2. The authors could improve their model validation techniques. Although I appreciate the use of interval-coverage and median relative error, most statisticians would prefer some use of “predictive scores” to evaluate model fit, as well as plots of model residuals. In a multivariate model, residuals can be shown via bivariate predictive ellipses, or via a Chi-squared statistic for each vector of residuals.

Please see response above on multivariate modeling. Figure 3 contains predicted vs. observed plots. Residual plots are below, and are also given as Supplementary Figure 1 in the SI.

3. The authors appear to use a variety of different methods including descriptive (sample correlation, Fig. 1), model-based (random-walk model for trait evolution, Fig. 1), and regression (phylogenetic-corrected least squares). However, there is relatively little comparison among methods, or illustration of how these methods could be combined in future studies.

We agree with the reviewer that more explicit comparison of methods is necessary to thoroughly interpret our results. We have added a paragraph of Discussion, below, in which we explain the seemingly counterintuitive result that phylogenetic signal-based models provide better predictions than life-history regression models even though regression models are found to explain significant variation in species nutrient content. One key idea, now emphasized in this section, is that sample size varies across methods, so that regression models are based on a smaller sample of species and their reduced predictive capacity may be a consequence of less precise parameter estimation.

“We find that most nutrients exhibit substantial phylogenetic signal—covariance among species in nutrient values is proportional to their shared evolutionary history—and a model

based solely on phylogenetic relationships and empirical estimates of phylogenetic signal provide reasonable predictions of species nutrient content. This phylogenetic signal-based model seems to provide better predictive ability than the phylogenetic regression model (Table 4) in spite of the fact that the latter model accounts for both phylogenetic signal and life history trait values. This seemingly counterintuitive result can be explained by two considerations. First, the sample size for estimating parameters of phylogenetic signal-based model is greater than that of phylogenetic regression model for each nutrient (see Tables 2 and 3). Some of the increased predictive capacity of the phylogenetic signal model is likely a consequence of better parameter estimation associated with a larger sample of species. Second, even though some life history variables (such as maximum depth) are significant predictors of some nutrients, the magnitude of effect is relatively small (Table 2). Therefore, inclusion of life history variables in nutrient predictions, as is done in the phylogenetic regression models, does not seem to offset the loss of sample size imposed by the additional requirement of having data on species life history. Note that we do not intend for our results to suggest that life history is unimportant in determining fish nutrient content, but rather that based on available data, incorporation of life history information does not improve predictions of species nutrient values. With the addition of more life history data, phylogenetic regression models may yield better predictions than the phylogenetic signal model. Also, particular clades of ray-finned fish may show stronger relationships between life history variables and nutrient content, though we were unable to investigate given limitations in available data.”

In addition, we have also added to our discussion of how these methods could be used or refined in future studies. We now clarify that our preference for the phylogenetic signal-based predictions is based on currently available data, but as more information on life history traits and improved phylogenies become available, we imagine that life-history regression models may further improve predictions.

“We identify several key steps in taking the conclusions of this research forward. First, improving the phylogenies of other classes important for human nutrition, including other marine seafood species as well as freshwater fish, is essential. Second, increasing the size of the nutrient validation sample—that is, measuring the nutrient content of more species directly—would enable a better utilization of the large amount of life history information available. It is possible, perhaps even likely, that the weak associations between life history variables and nutrient content in this study is due to an overly restrictive validation sample. We recommend expanding the sample by prioritizing nutrient assays in species that meet two criteria: (1) membership in families identified as potentially nutrient-rich, as presented in Table 5, and (2) availability of life history data, particularly maximum depth, trophic level, and maximum body size. Third—and approaching the small sample problem from the opposite direction—use of techniques like that of Thorson et al. (2017) to predict life history parameters would allow more existing nutrient information to be utilized.

“We utilize several methods in this paper: evolutionary correlations, Brownian-motion models of trait evolution, and phylogenetic least squares. We believe that the availability of data is the key determinant of which of these methods is preferable in predicting fish nutrient content. In the absence of life history information, evolutionary correlations and Brownian-motion

models provide a rough sense of which nutrients are likely to be found together in Actinopterygii. As phylogenies expand and nutrient and life history datasets grow, PGLS will likely improve on the predictive power of these methods. Overall, we believe that the combination of phylogenetic and life history information holds great potential for inexpensively inferring the nutrient content of a wide range of wild foods, and thereby quantifying the impacts of ecosystem transformation on human food and nutrition security.”

One option which would likely be useful is a phylogenetic method that includes missing values, where the vector of life-history parameters and nutrient contents would be analyzed jointly. Major axis regression would then show the “slope” between nutrients and life-history parameters, and nutrients could still be predicted for species with unknown phylogeny (i.e., based on the measured life-history traits for that species). Alternatively, major axis regression could be used for both the correlation matrix (Fig. 1), and the multivariate phylogenetic analysis (an extension of Eq. 1), and estimated slopes could be compared with regression results from the phylogenetic-corrected least squares method.

Please see response above to comment 1. We note also in the text that use of life history parameters in combination with phylogenetic relationships will likely improve predictive accuracy in the future; however, the current analysis—admittedly with a small sample, constrained by the availability of nutrient information—shows that life history parameters do not predict nutrients well.

Minor comments:

21 – My memory of references 2-3 is that they are forecasts of potential future changes. I believe there are other papers that are specifically focused on documenting past changes (Perry, Low, Ellis, & Reynolds, 2005; Pinsky, Worm, Fogarty, Sarmiento, & Levin, 2013, etc.)

Added both references suggested. We also wish to highlight future projections of change in the distribution and abundance of marine taxa, so also kept references 2-3.

25 – I think this paper documents “impacts in every marine ecosystem” but not “degradation”.

Changed “degrading” to “impacting”. We note that the primary motivation for this work is to develop tools for assessing nutritional impacts of threats to marine ecosystems.

78-79 – I don’t see any evidence that the variance in your data set is representative of fishes in general.

Sentence deleted.

107-110 – Again, these claims would be easier to quantify using major axis regression with the sample covariance. It appears that there is a trade-off between fat and protein. Using log-scaled variables, the ratio X of eigen-vector elements can be interpreted as “a 1% change in variable A is associated with a X% change in variable B”.

Please see response to comment #1 above.

128-131 – these would be interesting to compare with major axis regression results.

Please see response to comment #1 above.

135 – Please add “... when phylogenetic information are available.”

Added

176-177 – Please delete sentence and reference figure when explaining specific results.

Deleted; Figure 3 is referenced in earlier mention of error rates

226 – Please add “... when comparing results between a Brownian-motion model for trait evolution and a phylogenetic least squares approach”. I’m not convinced that you have explored the full suite of possible models, so an absolute statement of relative information between life-history and phylogeny is not supported.

Added. We note also that an exploration of life history-nutrient relationships with a larger sample may yield stronger results.

261 – This concern is exactly why it is beneficial to use a phylogenetic model that can handle missing values on both traits and nutrient-content samples. In this case, two unmeasured species with equivalent distance from the nearest measured relatives would still have their predictions updated by known life-history traits.

We agree, and we believe that, with a larger sample, life history traits would have proved more predictive of nutrient content. We believe that prediction of life history parameters using methods as in the reviewer’s paper, Thorson et al. 2017, would be a useful extension of this work. However, given currently available data, we were able to predict nutrient content for only a relatively small number of species in the Rabosky et al. (2013) phylogeny with sufficient life history information (protein, $n = 2$; total fat, $n = 2$; iron, $n = 8$; zinc, $n = 68$; vitamin A, $n = 83$; vitamin B12, $n = 89$; vitamin D, $n = 85$).

315 – I think you mean the residuals are expected to be normally distributed.

Changed

353 – What about covariance? There is clearly covariance among traits (Fig. 1). This can be informative and interpretable.

Please see response to comment #1 above. Although Fig.1 suggested strong covariance among traits, the PPCA scores did not prove valuable either in the PGLS or the phylogenetic prediction models.

Works cited

- Perry, A. L., Low, P. J., Ellis, J. R., & Reynolds, J. D. (2005). Climate Change and Distribution Shifts in Marine Fishes. *Science*, 308(5730), 1912–1915. doi:10.1126/science.1111322
- Pinsky, M. L., Worm, B., Fogarty, M. J., Sarmiento, J. L., & Levin, S. A. (2013). Marine taxa track local climate velocities. *Science*, 341(6151), 1239–1242.
- Thorson, J. T., Munch, S. B., Cope, J. M., & Gao, J. (2017). Predicting life history parameters for all fishes worldwide. *Ecological Applications*, 27(8), 2262–2276. doi:10.1002/eap.1606
- Warton, D. I., Wright, I. J., Falster, D. S., & Westoby, M. (2006). Bivariate line-fitting methods for allometry. *Biological Reviews*, 81(2), 259–291. doi:10.1017/S1464793106007007

Reviewer #3 (Remarks to the Author):

This study examines the evolution of nutrients relevant to human health found in major fish lineages of commercial importance. The study relies upon a previously published phylogeny of fishes (Rabosky et al., 2013) and literature sources of nutrient information for the comparative analyses and builds a novel predictive model of nutrient content for 7500+ ray finned fishes. I found the application of PCMs to nutrients and human health novel and fascinating. Although I think the study is missing some details about the analyses I think that in general the results are interesting, novel, and deserve publication. However I am not yet convinced that the story as presented here translates to a wide audience instead of, say, a journal more focused on fisheries. I agree that the phylogenetic associations reported for the evolution of the nutrients themselves are novel and that the tie in to human health and nutrition is potentially far-reaching. However right now I am missing justification, details, and possibly analyses that make an stronger case for publication in this journal.

We expand the argument in the opening section that such an approach is critical to understanding the global nutritional implications of changes in access to wild harvested fish. Such understanding is, in turn, a cornerstone of rational policy making to manage global fisheries to optimize nutritional outcomes. Lack of nutritional information is a critical impediment to such policy making—especially for designing nutritional interventions in developing countries—and expanding phylogenetic knowledge is a powerful tool to overcome this impediment.

“A lack of information about the nutritional composition of fish species prevents quantification of the nutritional threat to human populations of reduced consumption of wild-harvested fish. Measuring nutrient content is expensive and, as a result, nutrient analyses rarely capture the full breadth of vitamins, minerals, macronutrients, and fatty acids relevant to human nutrition. This information gap prevents the design

of rational fisheries management strategies and nutritional interventions to optimize public health outcomes in the face of rapidly changing marine conditions.

In this paper, we investigate the possibility that phylogenetic relatedness and life history information explain variation in the nutrient content of key fish species in the most commercially and nutritionally important class, Actinopterygii (ray-finned fishes). To our knowledge, this is the first time that such an approach has been explored. We then use the results of this analysis to develop a method for using shared phylogenetic history as a means of predicting the nutrient content of fish whose nutrient information has not yet been assessed, thus filling the information gap described above. Such predictions are especially critical in regions of the world with known nutrient deficiencies—often the same areas where laboratory capacity is limited. For fish specifically, recent years have seen an emerging desire among policymakers to design fisheries management and aquaculture development interventions with the specific goal of enhancing nutritional security.”

Here is a list of major and minor concerns...

- in Supplementary Tables 2 and 3 what is the link between these tables and the importance of fish to low income coastal populations?

SI Tables 2 and 3 shows that the fish included in the Actinopterygii dataset are important sources of the set of nutrients investigated. In addition, we add a reference to the Sea Around Us Project database showing that the majority of subsistence seafood consumption in low income countries comes from wild-caught ray-finned fish.

- is the list of species below SI Table 3 part of Table 3?

Yes; added header row to clarify

Why does this table contain non-actinopts (Meretrix, Ruditapes)?

Edited; initial research included non-Actinopterygii, was subsequently modified.

- what is known about the correlation of these nutrients in non-phylogenetically controlled studies? For example, the paper reports that 98% of species are rich sources of protein (and it looks like all are sources of protein). Under what conditions could a fish species not be a good source of protein? Are vitamins A and D known to co-occur in commonly fished species already? Is this association with fat noted as well?

To our knowledge, there are no large-scale analyses of nutrient correlations/clustering within fish species. The high protein content of nearly all fish species is well-known, and thus policy documents linking fisheries management and nutritional security focus on protein alone. However, our experience with fisheries experts is that there is a lack of understanding of micronutrients, vitamins and fats, and which species are particularly

rich sources for each of these. This study will help to broadly categorize sources of nutrition by taxonomic groups.

- are there any physiological, life history explanations for the evolutionary trends you report (lines 10-110)? Why can vitamin A concentration can evolve so quickly? Why are other traits slow to evolve?

The reviewer raises a good question. The low phylogenetic signal in vitamin A stands in contrast with the moderate to high phylogenetic signal in other nutrients. Although it is tempting to speculate on potential causes for this pattern, we avoid doing so because it is not possible to infer evolutionary process simply based on estimates of phylogenetic signal. High values of phylogenetic signal can be produced when the evolutionary rate is high or low as long as the evolutionary process is Brownian, which can occur under genetic drift or adaptive evolution when phenotypic optima shift by Brownian motion. In addition, low values of phylogenetic signal can occur under adaptive evolution when optima are fixed or when there are constraints on phenotypic evolution (Revell, LJ, LJ Harmon, and DC Collar. 2008, Systematic Biology 57: 591-601). So, the low phylogenetic signal in vitamin A does not imply a high evolutionary rate, only that evolution of vitamin A is non-Brownian.

- Figure 2 needs to be made much more accessible to a general readership. Designating the higher level taxonomy with common names of the groups (Scombriforms-tunas, mackerels, and allies as an example) would help readers understand the phylogenetic scope of the analysis. There should be a true timescale on the x axis. Representative illustrations of the major groups would also be helpful. These kinds of changes would help avoid the way the text currently directs the reader to results on the tree (lines 151-159 for example).

Changed. Added orders, common group names, a true timescale, and representative illustrations.

- Figure 3 yes your model allows you to reasonably predict nutrient content. However can you comment on the biological implications of width of the confidence envelope. For example in SI Table *Sebastes_taczanowskii* is within the model confidence limit for lipids but the predicted and measured values differ by an order of magnitude. Does this limit the utility of the phylogenetic prediction model?

Of the 1203 predictions we make (across 6 nutrients), 653 (54%) are within 25% of the observed value. Another 289 (24%) are within 50% of the observed value. Only 7 (>1%) differ by as much as an order of magnitude. Given the paucity of information on the nutrient content of fish species, we believe that this degree of inaccuracy is acceptable for population-level nutrient availability and intake accounting, and a major improvement over past estimates.

- Figure 3 can you say anything about the species that fall outside the model limits? Also, can you comment on why the model is consistently underestimating the nutrient content in its predictions?

We did not detect any pattern in the species that fall outside of the prediction intervals. These species are dispersed across major lineages of ray-finned fish and different sets of species fall outside of prediction intervals for different variables. Although we did not detect a bias in our predictions (mean deviation between predicted and measured variables is nearly zero), we suspect that the reviewer has noticed that all species falling outside prediction intervals have measured values greater than predicted. This effect is a consequence of a lower bound on values falling below the lower confidence limit; if the lower confidence is near zero, then there is little room for measured values to fall below the interval.

This comment spurred us to clarify in the manuscript that the width of confidence intervals is proportional to the branch length separating an unmeasured species from the most closely related measured species.

- I could not find the data file for the 7500+ species that are predicted (apologies if it was included but I could not find it in the SI).

Attached

- 148-49 citation for source(s) of data?

Sources are listed in references 21-33 in the Methods section, as well as the caption to SI Table 2. This is now noted in the lines to which the reviewer refers.

- 1116 this strikes me as quite vague. If observed correlations do indeed predict unmeasured nutrients then this extends the reach of the paper. Are there any citations here that support your statements? Are there any subsets of the data in hand that could demonstrate the ability of your analyses to predict levels of unobserved nutrients?

We perform a validation exercise below in the paper to test the accuracy of predictions; see Tables 4-5, and Figure 3 particularly, as well as accompanying discussion.

- The authors should provide a data file with the raw values of the nutrient levels and a citation to the source of the data.

Data file with the dataset used is included. Citations are given in SI Table 2's caption, and the associated reference list.

- I would like to see the SI include the tree file, data files, and scripts used to conduct the analyses so that they may be repeated.

All materials above now included:

Tree file: “Rabosky_etal2014.nwk”

Data: “PhyloFishNutrition.csv”

*Scripts: “evolutionaryCorrelations.R”, “phylogeneticSignal.R,”
“phylogeneticLeastSquares.R,” “phylogeneticPrediction.R”*

ⁱ Golden, C.D. et al. Fall in fish catch threatens human health. *Nature* **534**, 317–320 (2016).

ⁱⁱ Thilsted et al. 2016

ⁱⁱⁱ Béné et al. 2016

^{iv} Cisneros-Montemayor et al. 2016

^v Mozaffarian and Rimm 2006

^{vi} Lund 2013

^{vii} Food and Agricultural Organization (FAO) of the United Nations. 2017. FishStatJ: Software for Fishery Statistical Time Series. In: *FAO Fisheries and Aquaculture Department* [online]. Accessed 13 July 2017 from <http://www.fao.org/fishery/statistics/software/fishstatj/en>

Reviewers' comments:

Reviewer #1 (Remarks to the Author):

The authors have done a good job of incorporating the feedback from the reviews. From a nutrition perspective, there are a few minor outstanding issues that need to be addressed prior to publication:

-Nutritional content/intake should be revised to nutrient content/intake throughout the text.

-Micronutrients should be changed to nutrients (given that fatty acids are macronutrients) in the sentence that states: "The consequences of these changes for human health, and especially for nutritional intake, are likely to be severe because fish provide critical micronutrients essential to human nutrition, including iron, zinc, vitamin A, vitamin B12, omega-3 and omega-6 fatty acids, and others."

-"Ray-finned fish—the class Actinopterygii—are particularly important to low-income populations, as suggested by the composition of subsistence wild-caught intake in developing countries.": Can the authors be more specific about the composition? It's still not clear to the reader unless they look up the reference provided.

-Macronutrients and fatty acids are listed separately in some cases but fatty acids are a macronutrient. It may be superfluous to include both.

-"These nutrients are generally present in relatively high concentrations in seafood—although, as we describe below, not all the above nutrients are present in appreciable quantities in all species—and are critical for human nutrition.": what is appreciable quantities? This is a bit vague.

Reviewer #2 (Remarks to the Author):

Major comments (from James Thorson)

In these revisions, the authors have made a few changes in the main text in response to my prior comments. In particular, I previously highlighted the potential benefit of major axis regression to interpret results. In the response-to-reviewers file, the authors have shown that conducting a phylogenetic analysis on a PCA variable extracted from their data does not offer additional insights beyond those in the paper. I am satisfied by this response regarding major axis regression, although I am politely skeptical that it represents the final word on how best to combine phylogenetic and life-history information in this data set (as the authors also acknowledge). I also thank the authors for making the data set publicly available, which I believe will be a useful contribution to fisheries science in itself.

However, I also believe that the authors have failed to fully address the "missing data modelling" issues that I highlighted previously. There are several ways to implement missing-data models that would simultaneously impute missing values (under the assumption that they are missing at random) while implementing the existing regression and/or phylogenetic model. As noted in my previous review, the FishLife model from Thorson et al. (2017) does data imputation, phylogenetic, and life-history regression simultaneously, and by doing so is able to predict missing values for all described fishes. There are several places where the authors should continue to revise the text to avoid incorrect statements, e.g., that regression-based methods require knowing life-history parameters for all species (line 216). I also hope that the authors will correct their analysis so that all methods fit to the same full data set, while using missing-data methods to allow for statistical comparison of model performance.

Minor comments

- It seems plausible that environmental temperature is a major driver of the variation that is currently attributed to body size and depth, given that temperature is predicted under the metabolic theory of ecology to drive growth and is clearly associated with depth. Is there any reason not to include

temperature measures available from FishBase?

- line 208 when adding line numbers on track-changes version: Please state your use of 95% confidence intervals when interpreting the 8.2% exceedance proportion in the main text.
- line 216: This statement that the regression approach requires a complete data set for all species is not true, as illustrated e.g. by Thorson et al. (2017)
- Table 4: Instead of error rate (which is unknown given that you don't know the truth), I would list 100% minus the listed value and label that "95% confidence interval coverage"
- 290-296: I agree with the authors that the different data set used in each analysis presented in this study makes it hard to compare results among analyses and instead means that the model-comparison is tied up with a discussion of the different data set used by each method. However, this problem would go away if you use a missing-data model, where missing data are predicted as random effects under a standard "data augmentation" approach. This would then allow for statistical measures of model fit and would put all models on the same footing.

Reviewer #3 (Remarks to the Author):

The authors have made substantial changes to the paper and now make a much stronger case for publication in the journal. The analyses are much clearer. I don't care for the way Fig 3. is arranged as it is difficult to connect the taxonomic information with the topology itself since the names are below the heatmap and I would encourage the authors to try other arrangements. However my overall assessment is that the authors have done a sufficient job to address my concerns from the prior round of review and I would be happy to published.

Responses to reviewers (#2)

Reviewer #1 (Remarks to the Author):

The authors have done a good job of incorporating the feedback from the reviews. From a nutrition perspective, there are a few minor outstanding issues that need to be addressed prior to publication:

-Nutritional content/intake should be revised to nutrient content/intake throughout the text.

Changed

-Micronutrients should be changed to nutrients (given that fatty acids are macronutrients) in the sentence that states: "The consequences of these changes for human health, and especially for nutritional intake, are likely to be severe because fish provide critical micronutrients essential to human nutrition, including iron, zinc, vitamin A, vitamin B12, omega-3 and omega-6 fatty acids, and others."

Changed

-"Ray-finned fish—the class Actinopterygii—are particularly important to low-income populations, as suggested by the composition of subsistence wild-caught intake in developing countries.": Can the authors be more specific about the composition? It's still not clear to the reader unless they look up the reference provided.

Now included; using SAUP data, we calculate that 80.6% of the tonnage of landed marine fish in the subsistence sector in 2010 belong to the class Actinopterygii.

-Macronutrients and fatty acids are listed separately in some cases but fatty acids are a macronutrient. It may be superfluous to include both.

Changed

-"These nutrients are generally present in relatively high concentrations in seafood—although, as we describe below, not all the above nutrients are present in appreciable quantities in all species—and are critical for human nutrition.": what is appreciable quantities? This is a bit vague.

Reworded to *"These nutrients are generally present in relatively high concentrations in seafood—although, as we describe below, not all species are important sources of the all the above nutrients—and are critical for human nutrition."* We refer here to the FAO designations of "sources" and "rich sources" of nutrients, defined in Supplementary Table 1 and referenced in the subsequent paragraph.

Reviewer #2 (Remarks to the Author):

Major comments (from James Thorson)

In these revisions, the authors have made a few changes in the main text in response to my prior comments. In particular, I previously highlighted the potential benefit of major axis regression to interpret results. In the response-to-reviewers file, the authors have shown that conducting a phylogenetic analysis on a PCA variable extracted from their data does not offer additional insights beyond those in the paper. I am satisfied by this response regarding major axis regression, although I am politely skeptical that it represents the final word on how best to combine phylogenetic and life-history information in this data set (as the authors also acknowledge). I also thank the authors for making the data set publicly available, which I believe will be a useful contribution to fisheries science in itself.

However, I also believe that the authors have failed to fully address the “missing data modelling” issues that I highlighted previously. There are several ways to implement missing-data models that would simultaneously impute missing values (under the assumption that they are missing at random) while implementing the existing regression and/or phylogenetic model. As noted in my previous review, the FishLife model from Thorson et al. (2017) does data imputation, phylogenetic, and life-history regression simultaneously, and by doing so is able to predict missing values for all described fishes. There are several places where the authors should continue to revise the text to avoid incorrect statements, e.g., that regression-based methods require knowing life-history parameters for all species (line 216). I also hope that the authors will correct their analysis so that all methods fit to the same full data set, while using missing-data methods to allow for statistical comparison of model performance.

We thank the reviewer for the suggestions, and for pointing us towards the FishLife model and package. We investigated various issues relevant to this topic. Our comments:

1. **Rationale for predictor choice.** First, we note that the selection of our predictors (depmin, depmax, troph, etc.) was guided by both conceptual importance—particularly ecological factors—and data availability, i.e., minimizing the number of missing values. To provide an example relevant to another of the reviewer’s suggestions: we excluded temperature because only 16% of our included species had max/min temperature data; most of our dataset would need to be imputed for inclusion.
2. **Constraints to modeling sample size: outcomes vs predictors.** We also note that the most serious constraints to sample size in our models are *outcome* variables, not predictors:

Type	Variable	valid	missing
Outcomes	protein	370 (98.7%)	5 (1.3%)
	fat	367 (97.9%)	8 (2.1%)
	omega_3	106 (28.3%)	269 (71.7%)
	omega_6	106 (28.3%)	269 (71.7%)
	iron	343 (91.5%)	32 (8.5%)
	zinc	177 (47.2%)	198 (52.8%)
	vitA	146 (38.9%)	229 (61.1%)
	vitB12	122 (32.5%)	253 (67.5%)
	vitD	122 (32.5%)	253 (67.5%)
Predictors	habcat	370 (99.7%)	1 (0.3%)
	depmin	256 (69.0%)	94 (31.0%)
	depmax	246 (66.3%)	125 (33.7%)
	bioca	365 (98.4%)	6 (1.6%)
	maxlen	366 (98.7%)	5 (1.3%)
	lencat	367 (98.9%)	4 (1.1%)
	a_lw	329 (88.7%)	42 (12.3%)
	b_lw	329 (88.7%)	42 (12.3%)

We see here that 6 of the 9 outcome variables have more than 50% data missing, which—if outcomes are not imputed—results in differing model samples.

- Imputation of outcomes?** After some investigation, we chose not to impute the nutrient content outcome variables, for several reasons:
 - For many cases, imputation of missing observations would rely only on protein, total fat, and iron content as predictors. Bivariate correlations (see Figure 1 in ms), as well as multivariate models we ran for this exercise, suggest that such a model predicts zinc, vit A, vit B12, and vit D poorly. For example, here is a linear model predicting zinc (others available on request):

```
Call: lm(formula = zinc ~ protein + fat + iron, data = bioca)
```

Coefficients:

	Estimate	Std. Error	t value	Pr(> t)
(Intercept)	1.039464	0.318887	3.260	0.00134 **
protein	-0.027522	0.015964	-1.724	0.08650 .
fat	-0.007157	0.009675	-0.740	0.46047
iron	0.390722	0.040700	9.600	< 2e-16 ***

Signif. codes: 0 '***' 0.001 '**' 0.01 '*' 0.05 '.' 0.1 ' ' 1

Residual standard error: 0.5872 on 172 degrees of freedom
(199 observations deleted due to missingness)

Multiple R-squared: 0.3633, Adjusted R-squared: 0.3522

F-statistic: 32.72 on 3 and 172 DF, p-value: < 2.2e-16

- Given the poor explanatory power of these models, we hesitate to use taxonomic (or phylogenetic) structure to deal with residuals on these models; because of the

heavy reliance on taxonomy/phylogeny, many species would have nearly the exact same imputed nutrient content, i.e., relationships would be over-identified. Such reduced variance in the outcome variable complicates interpretation of the modeling results currently included in the paper.

4. **Other predictors using FishLife.** However, FishLife's missing-data functionality offers the opportunity to test other predictors of nutrient content. We used FishLife to obtain and predict values for the included 7 life history variables and temperature, for all 371 species in our dataset:
 - L_{∞} : length at infinity (asymptotic length)
 - K : growth rate (1/year) at which asymptotic length is approached
 - W_{∞} : asymptotic weight
 - t_{\max} : maximum age of the population
 - t_m : age at first maturity
 - M : natural mortality
 - L_m : length at maturity
 - Temperature: mean environmental temperature expressed as $1000 / (C + 273.15)$

We then ran PGLS models with these predictors. The results and code are given in the attached folder "*PGLSImputed*", which contains the following files:

- *bionut.PGLS_imputed.R*: code for running the PGLS regressions
- *imputedLH_cormatrix.csv*: phylogenetic correlation matrix using the new predictors
- *nutPGLS_imputed_fitTable.csv*: goodness of fit tables for the various nutrient models (using new predictors)
- *nutPGLS_imputed_LRT1.csv* and *nutPGLS_imputed_LRT2.csv*: likelihood ratio tests for models
- *nutPGLS_imputed_table.csv*: regression results for each model

To summarize, we found that the length at infinity variable is a significant predictor of iron and zinc, and temperature is a significant predictor of fat and Vitamin A. We find no other significant regression coefficients. We note again that sample size is strongly constrained by outcome variables.

Given these results, we chose to examine whether the new set of predictors, with imputed values, perform in the same prediction validation exercise we performed with the original set of variables (as in Table 4 in the manuscript). The results are included in the *phyloPredValidPGLS_imputed.error* file. We find that the new variables perform slightly better than the original set of variables, but do not outperform phylogenetic signal alone.

5. **Overall conclusions.** We were then faced with the choice of revising the manuscript using the imputed dataset, or retaining the original dataset. The imputed dataset offers a major advantages: 1) It allows models to be compared, as all contain the same species list; and 2) the new set of predictors performs slightly better than the original set.

However, we chose to retain the original set of predictors, for the following reasons:

- The use of taxonomy rather than phylogeny to deal with residuals in the imputation exercise would be inconsistent with the rest of the manuscript. It may indeed be the case that taxonomic relationships approximate phylogenetic relationships for a given sample, but given that we wish for this technique to be generally deployable for *any* sample, we prefer the use of phylogenetic information.
- The original set of predictors includes a variety of ecological variables, whereas the new set focuses more closely on population dynamics. Regardless of the PGLS results, we believe there is merit in examining the relationship between ecological variables and nutrient content.
- Many observations in the new predictor dataset are imputed. We believe that the use of so many imputed values requires a further uncertainty analysis with the PGLS modeling exercise, which we believe is outside the scope of this paper.

Minor comments

- It seems plausible that environmental temperature is a major driver of the variation that is current attributed to body size and depth, given that temperature is predicted under the metabolic theory of ecology to drive growth and is clearly associated with depth. Is there any reason not to include temperature measures available from FishBase?

We searched FishBase for TempMin and TempMax. These temperature data are implemented at the stock level, not the species level; of the 564 stocks (spanning 390 species), only 88 (15.6%) had TempMin data and 90 (16.0%) had TempMax data. We therefore chose to exclude temperature, given that most of our dataset would need to be imputed for inclusion. See comment above for further discussion.

- line 208 when adding line numbers on track-changes version: Please state your use of 95% confidence intervals when interpreting the 8.2% exceedance proportion in the main text.

Changed to *“observed nutrient values fall within the prediction 95% confidence intervals for at least 8.2% of cases across all variables”*

- line 216: This statement that the regression approach requires a complete data set for all species is not true, as illustrated e.g. by Thorson et al. (2017)

Changed to *“Moreover, the phylogenetic prediction method can be readily applied to any species whose phylogenetic relationship to measured species has been estimated. Regression-based prediction requires this information in addition to information on species life history, although imputation methods may help to predict missing observations from a combination of other life history predictors and taxonomic/phylogenetic information (Thorson et al. 2017).”*

- Table 4: Instead of error rate (which is unknown given that you don't know the truth), I would list 100% minus the listed value and label that “95% confidence interval coverage”

Changed

- 290-296: I agree with the authors that the different data set used in each analysis presented in this study makes it hard to compare results among analyses and instead means that the model-comparison is tied up with a discussion of the different data set used by each method. However, this problem would go away if you use a missing-data model, where missing data are predicted as random effects under a standard “data augmentation” approach. This would then allow for statistical measures of model fit and would put all models on the same footing.

Please see initial response above

Reviewer #3 (Remarks to the Author):

The authors have made substantial changes to the paper and now make a much stronger case for publication in the journal. The analyses are much clearer. I don't care for the way Fig 3. is arranged as it is difficult to connect the taxonomic information with the topology itself since the names are below the heatmap and I would encourage the authors to try other arrangements. However my overall assessment is that the authors have done a sufficient job to address my concerns from the prior round of review and I would be happy to published.

Modified; we added gray boxes on the heatmap figure to better connect illustrations of taxonomic groups with phylogenetic and nutritional information.

Additional note to reviewers: Table 5 has been slightly revised from the last version. We now considered a species nutrient rich if its nutrient value fell within the top 5% of species nutrient values; previously we considered species nutrient rich if they fell within the largest 500 species values. Also, the table now includes only families with 10 or more nutrient rich species.

REVIEWERS' COMMENTS:

Reviewer #2 (Remarks to the Author):

In these revisions, the authors have made useful edits in response to comments from three reviewers. Most substantially, they have included in their response-to-reviewers some additional analysis regarding the use of imputed life-history data to predict nutrient contents, which I had predicted could be useful on the basis that metabolic theory would predict some covariance between fat reserves and life-history traits/parameters. I believe that the authors' response regarding the tradeoff between additional complexity (requiring additional exploration) vs. their simplified analysis is appropriate, and believe the additional text is a useful nod towards this potential future research. While I do not full agree with the authors' response (e.g., the implication that it is not feasible to use data - imputation for response variables, or that low sample sizes is a major obstacle), I believe that the current draft represents a very useful contribution to the literature, and recommend that it be published as is.